# Investigation of complex coastline geometry impact on the evolution of storm surges along the east coast of India: A sensitivity study using a numerical model

Pawan Tiwari[1*], Ambarukhana D. Rao[1], Smita Pandey[1] and Vimlesh Pant[1]

[1]Centre for Atmospheric Sciences, Indian Institute of Technology Delhi, India.

*Correspondence to*: Pawan Tiwari (ptiwari474@gmail.com)

**Abstract.** A comprehensive investigation is made on the generation of storm surges along the coast in response to complex coastline geometry using a standalone advanced circulation (ADCIRC) model. The study deals with sensitivity experiments by using various idealized concave/ convex model domains with the same intensity of parallel cyclone tracks. It demonstrates that a sharp curvature along with the landfall location of each track within the domain has more influence on the surge evolution. Peak surges (PS) are generated in the domain for the tracks possessing strong onshore winds, while intense alongshore winds are responsible for PS spreading along the coast. The time evolution of both positive and negative surges along the coast is also demonstrated. The propagation of energy density per unit length associated with surge waves is computed for concave coasts to explain the funnelling effect. Development of PS is also seen with actual intricate coastal stretch having concave and convex coastlines along the east coast of India using parallel tracks, which is consistent with that of idealized experiments. Further simulations are carried out along real coastlines with different approach angles of the track exhibit that maximum PS are not always aligned to the east of the track. Depending on track angle, PS may also develop on the west side. Experiments on real coastlines also indicate that PS in the concave coastline is more influenced by cyclone's radius of maximum winds. Surge generation with different radius of maximum winds is seen to the west side of the track though the cyclone is not making a landfall in this region. Simulation with a recent cyclone, Michaung reveals that occurrence of storm tides is also seen to west of the landfall as the track moves parallel to the coast, which agrees with the observations. This study signifies importance of local coastline configuration, particularly in the concave coasts, on the intensification of storm surges.

## 1 Introduction

The shape of the coastline, whether concave or convex, significantly influences the generation and propagation of storm surges (SS) during any extreme weather events such as tropical cyclones (TC). Concave coastlines are curved inward, which tends to focus and funnel SS waters into a narrower channel, amplifying surge heights and their impact on the coastal regions. The concave coastline forces the storm waters to converge as any cyclone approaches the coast, resulting in a buildup of water levels that can cause more severe flooding near coastal areas (Sebastian et al., 2019). On the other hand, convex coastlines, which protrude outward, can help dissipate the energy of incoming SS by spreading the water over a wider area. The associated dispersion may reduce the SS's intensity, mitigating its impact on coastal communities. Hence, concave coastlines amplify SS heights, which may pose a greater risk to coastal populations and infrastructure. In contrast, convex coastlines serve as natural buffers, dispersing and reducing the impact of SS (NOAA, 2022).

There have been many studies in the past on the generation of SS along the coast, which depend on many factors, such as cyclone intensity (Zhong et al., 2010; Resio and Westerink, 2008), forward speed (Rego and Li, 2009), $R_{max}$, known as the radius of maximum winds (Peng et al., 2004, 2006), orientation of the coastline (Dube et al., 1982), coastal bathymetry (Johns et al., 1983), width and slope of the continental shelf (Poulose et al., 2017), and angle at which the cyclone is crossing the coast (Zhang, 2012). Several researchers developed earlier many numerical models to realize the contribution of various factors in the development of SS in the Bay of Bengal (BoB) (Das, 1972; Johns et al., 1983; Dube et al., 1994; Rao et al., 1997; Murty and Kolukula, 2023; Pandey and Rao, 2018; Pandey et al., 2021; Tiwari et al., 2024).

Ueno (1981) developed a simplified model, investigated the effect of SS over a curved coastline, and learned that even slightly curved shoreline produces more SS than straight coasts. According to Dassallas and Lee (2019), regions with concave coastlines are prone to SS much higher than the straight coastline of the same topography and bathymetry. It is further revealed that the flood from SS is more likely to occur if the typhoon approaches perpendicularly than to that of the parallel coastline. Xuan et al. (2021) discussed that the SS enhancement is highly dependent upon coastline shape, with flaring (convex) coastlines suffering from higher SS mostly when the TCs strengthen into severe cyclones. While for the arch (concave) coastline, the higher surge enhancement occurs during severe or super typhoons. Hope et al. (2013) found that the translation speed and coastline geometry significantly influence maximum SS heights, with slender bays showing increased susceptibility to extreme SS from fast-moving hurricanes, while short and wide bays exhibited elevated SS levels during slower storms. According to Qian et al. (2024), the shape of the coast is majorly represented by the bay category, leading to high maximum SS in narrow bays when a TC strikes. For instance, longer and broader bays are prone to higher SS during slow TC events. Under certain conditions, square bay cases showed utterly different behavior compared to open-coast cases, and as a result, maximum SS was lowered. Subramaniam et al. (2019) reported that in the case of convex dikes, the wave run-up is increased at the center as the opening angle increases. Contrary to this, for concave dikes, wave run-up enhances as the opening angle

gets narrower. As studied by Eurotop (2018), the concave curvature may contribute to the accumulation of wave energy, enhancing the wave's run-up and overtopping. The opposite is the case for convex curvature because the distribution of wave energy affects the wave run-up, and overtopping will consequently decrease.

The shape of the coastline is one of the factors which determines the vulnerability of the region due to tsunami and SS waves. The study of Zhao and Niu (2022) highlighted that when a tsunami wave approaches a concave coastline, it meets with the uniqueness of an environment that irregularly concentrates and redirects the wave's energy. The waves are tightly squeezed in the concave coast, which creates a natural amplifier that makes them higher as they approach the center of the coastline. Salaree et al. (2021) also highlighted how the coast's concave geometry concentrates the tsunami wave amplitudes with idealized bathymetry, consistent with edge wave theory (Munk et al., 1956).

Sebastian et al. (2019) examined the concave coast features on SS generation through net-flux movement computation. A concave-shaped coastline, along with the cyclonic approach angle defined as the angle between the tangent drawn at the landfall location and the cyclone track measured clockwise, leads to an impact within the concave coast. This contributes to the complicated behavior of non-linear surges in the sites situated inside the concave coast among the entire coastlines. Pandey and Rao (2019) performed a numerical assessment of approaching hurricane tracks at different angles. They unveiled that the higher SS and its interactions with tides and wind waves are associated with the perpendicular cyclone track. Irish et al. (2008) employ numerical simulations to quantify the effect of storm track variation on SS in different slope areas along the Gulf of Mexico as the northern gulf boundary has a linear coastline orientated east-west. Numerical simulations of Flierl and Robinson (1972) for the northern part of BoB demonstrated that surge heights depend on the track angle between two perpendicular straight beaches.

Earlier studies concerned only with the SS development along the actual concave coastlines, and mostly deal with the short waves generated by the normal winds. However, no mention is made about impact of any irregular coastal stretch associated with the cyclonic winds with different approaching angles on the generation mechanism of SS. Limited studies were caried out with concave coastlines but none with convex coastlines. Also, past studies suggest the east side of the cyclone landfall is affected mostly due to higher onshore winds whereas the west side experiences less damage due to offshore winds. However, these studies do not account for the surge behavior with the changing curvature for both the concave and convex shapes. A detailed analysis regarding surge generation and its mechanism in the concave/convex shaped coastlines approaching is needed for a better understanding of the development of SS, which is the focus of this study.

The coastline along India's east coast covers about 2500 km with most irregular geometries like concave, convex, and straight coastlines of varying horizontal dimensions shown in Fig.2. Several cyclones in the past hit these coastal stretches with different approach angles like Lehar (2013), Helen (2013), Laila (2010) and Ogni (2006) cyclones. Some of the cyclones move parallel to these areas without making any landfall, for example Roanu (2016), Jawad (2021), and 1977 Andhra Pradesh

cyclones. They can generate significant SS and associated coastal inundation, particularly on the west side of the track. Hence, it emphasized the influence of complex coastal geometry and cyclone approach angle on the generation of SS. Owing to its significance, the present study thoroughly investigates the role of different coastal shapes and approach angles (parallel and oblique) in generating SS over idealized and natural coastlines.

The study examines the evolution of SS by considering idealized parallel tracks based on the information from the historical cyclone track data over the BoB. The study uses the advanced circulation (ADCIRC) model to analyse peak surge (PS) trends over various dimensions of concave/convex coasts. The experiments are divided into three sections; the first one computes and validates storm tides (ST) for the recent cyclone, named Michaung (RSMC report, 2023), making landfall in one of the concave regions of the east coast of India. The second experiment is sensitivity analysis based on idealized and different forms of concave/convex shorelines along with parallel cyclone tracks landfalling in these domains. The third part considers the actual shoreline representing the east coast of India by taking oblique, perpendicular, and parallel tracks concerning the landfall location. The present study investigates the surge dynamics, mechanism, and propagation along the coast for idealized/real concave/convex coastlines, which is crucial for effective coastal planning, hazard mitigation, and disaster preparedness efforts in vulnerable regions.

## 1.1 Synoptic History of the Michaung Cyclone

The Michaung cyclone initially formed as cyclonic circulation in the neighbouring south Thailand on 26th November 2023. Following a westward path by 1st December, it was strengthened into a depression over the southeast BoB as shown in Fig. 1. The cyclone continued its north-westward path till 4th December, off the coastlines of north Tamil Nadu and south Andhra Pradesh. It was strengthened further into a severe cyclonic storm (SCS) over the west-central and adjacent southwest BoB. Later, it made landfall as SCS near Bapatla in the afternoon of 5th December with maximum sustained winds of 90-100 km/h. It was then moved nearly northward, parallel close to the coast of south Andhra Pradesh. In the North Indian Ocean, the Michaung was the sixth cyclone that formed in the same year, with maximum sustained winds surpassing 55 knots (101 km/h). A lot of flooding was observed around Chennai due to cyclone-induced precipitation and intense gale winds that hit the coastal regions of south Andhra Pradesh and north Tamil Nadu. A maximum SS of about 1.0 -1.5m was observed and flooded low-lying parts of south coastal Andhra Pradesh. The entire cyclone track covers a long distance of about 1150 km (RSMC report, 2023).

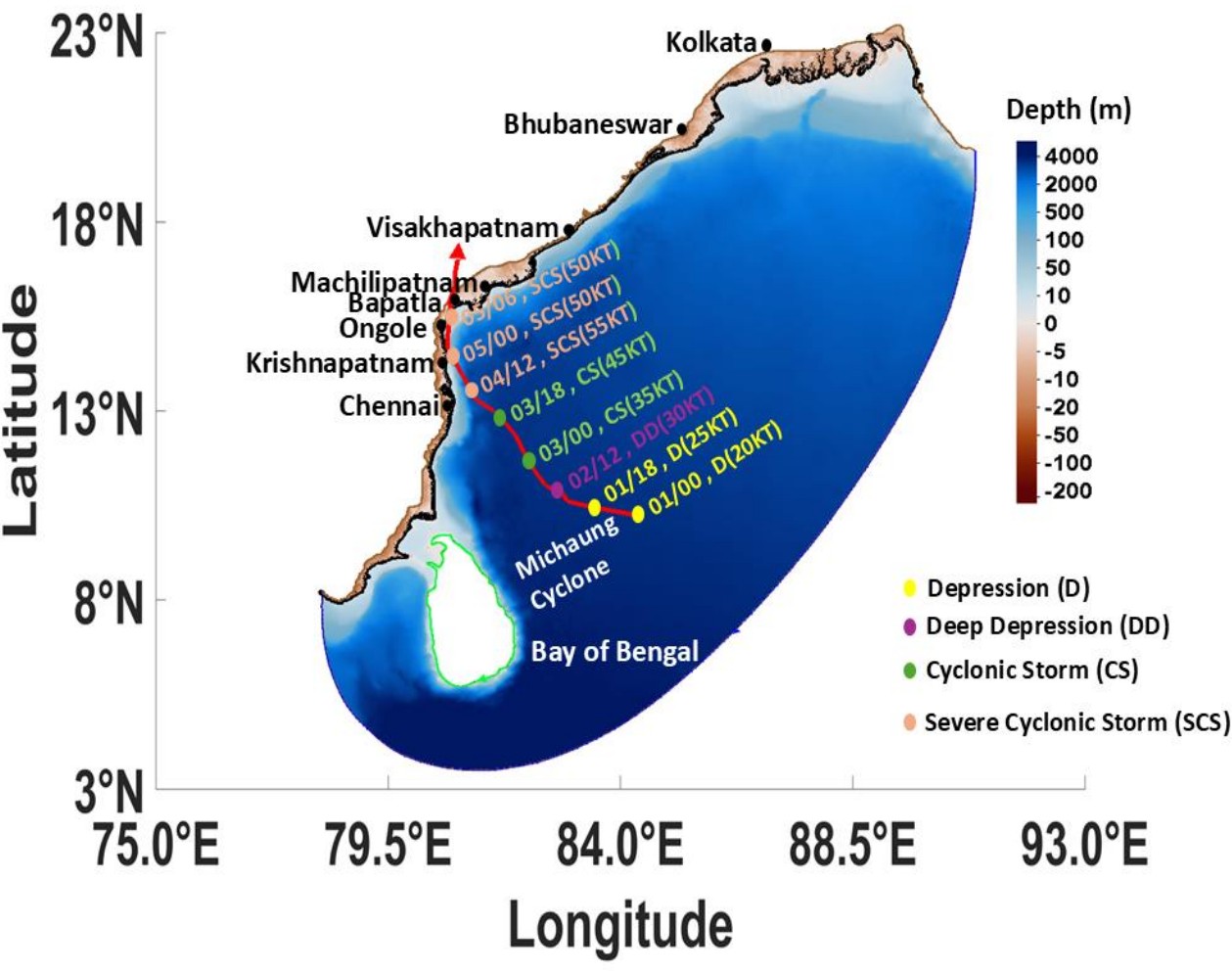

**Figure 1. Michaung cyclone track and its intensity with time.**

## 2 Model Description

### 2.1 Hydrodynamical surge model

The study utilizes the 2D depth-integrated hydrodynamic Advanced Circulation (ADCIRC-2DDI) model developed by Luettich et al., 1992 and Kolar et al., 1994a. Luettich et al. (1992) provided a detailed explanation of the ADCIRC-2DDI hydrodynamic model. More information about the model can be found at (https://adcirc.org/home/documentation/users-manual-v50/). The momentum and continuity equations of the ADCIRC-2DDI model are solved on an unstructured grid using the Galerkin finite-element method. The model solves the generalized wave continuity equation (GWCE) to determine the

maximum water elevation and the vertically integrated momentum equation for the currents. The model's governing equations are approximated using Boussinesq, incompressibility, and hydrostatic pressure. The ADCIRC incorporates boundary conditions such as normal flux per unit width and free-surface elevation from different tidal components at the open boundary.

The model employs a hybrid bottom friction formulation with a minimum drag coefficient of 0.0015 to compute SS, the chosen value is due to model stability. The duration of the ramp function for all simulations is around one day. The weighting factor ($\tau_0$) in GWCE is 0.05, according to Pandey and Rao (2018), while the eddy viscosity is 2 $m^2$/s, as reported by Cyriac et al. (2018). $\tau_0$ weights the relative contribution of the primitive and wave portions of the GWCE. The Ramp function simply scales the applied forcing, making it a unit less value that varies from 0 (no forcing) to 1 (full forcing). The detailed formulation for the hybrid friction is mentioned in Luettich et al. (1992). The drag coefficient, Ramp function, and weighing factor are dimensionless. The simulations utilize the wind drag formulation proposed by Garratt (1977).

## 2.2 Holland wind module

Wind stress and horizontal variations in surface air pressure primarily produce SS. The inverse barometric effect results from the impact of changing air pressure on the sea surface. The sea level rises one cm for each hPa decrease in atmospheric pressure. The wind stress on the sea surface leads to a notable fluctuation in sea level, influenced by the maximum wind speed and geographic distribution of winds (Ross, 1854). The study uses a cutting-edge wind module (Holland et al., 2010, cited as H10) as an input in the model to compute horizontal cyclonic winds. H10 is a revised, more advanced version of the earlier model by Holland (1980) known as (H80) that uses an improved estimation of the wind speed and wind direction near the cyclone eye, optimizes the information that is essential to determine the $R_{max}$, resolves bimodal wind profiles and provides higher spatial resolution for a more precise assessment of the wind parameters in TCs. Murty et al. (2020) used the H10 wind module in the hydrodynamic and wave model in the BoB.

## 3. Data and Methodology

### 3.1 Experiments using cyclone Michaung (Exp1)

In the first set of experiments (Exp1), the recent cyclone of Michaung is considered, which made landfall in the same region. The information for the cyclone is taken from the IMD best track data (https://rsmcnewdelhi.imd.gov.in/report.php?internal_menu=MzM=). The INCOIS website provides the tide-gauge data used in this study (https://incois.gov.in/portal/datainfo/drform.jsp). In Exp2 and Exp3, the tides are not considered in the model; however, in Exp1, the tides are simulated in the model to compute storm tides (ST) during the Michaung cyclone period. The model uses 13 tidal constituents (K1, M2, N2, O1, P1, S2, K2, L2, 2 N2, MU2, NU2, Q1, and T2) at the open boundary, extracted from the "Le Provost" tidal database (Le Provost et al., 1998) for the simulations of tides. With these configurations, model simulations are performed with a standalone ADCIRC model to compute SS. However, we are interested in plotting maximum SS at any location during the cyclone period, denoted here as PS. Among all PS values along the coast, the maximum PS is referred to as MPS.

## 3.2 Experiments using idealized coastline (Exp2)

The east coast of India has a very irregular coastline consisting of concave, convex, and straight coastlines, as shown in Fig. 2. To understand the effect of concave, convex, and straight coastlines on the generation of SS, the concave-shaped coastline near Machilipatnam is selected for the study as this area is frequently affected by the SS. The bathymetry of the region also follows a concave-shaped coastline.

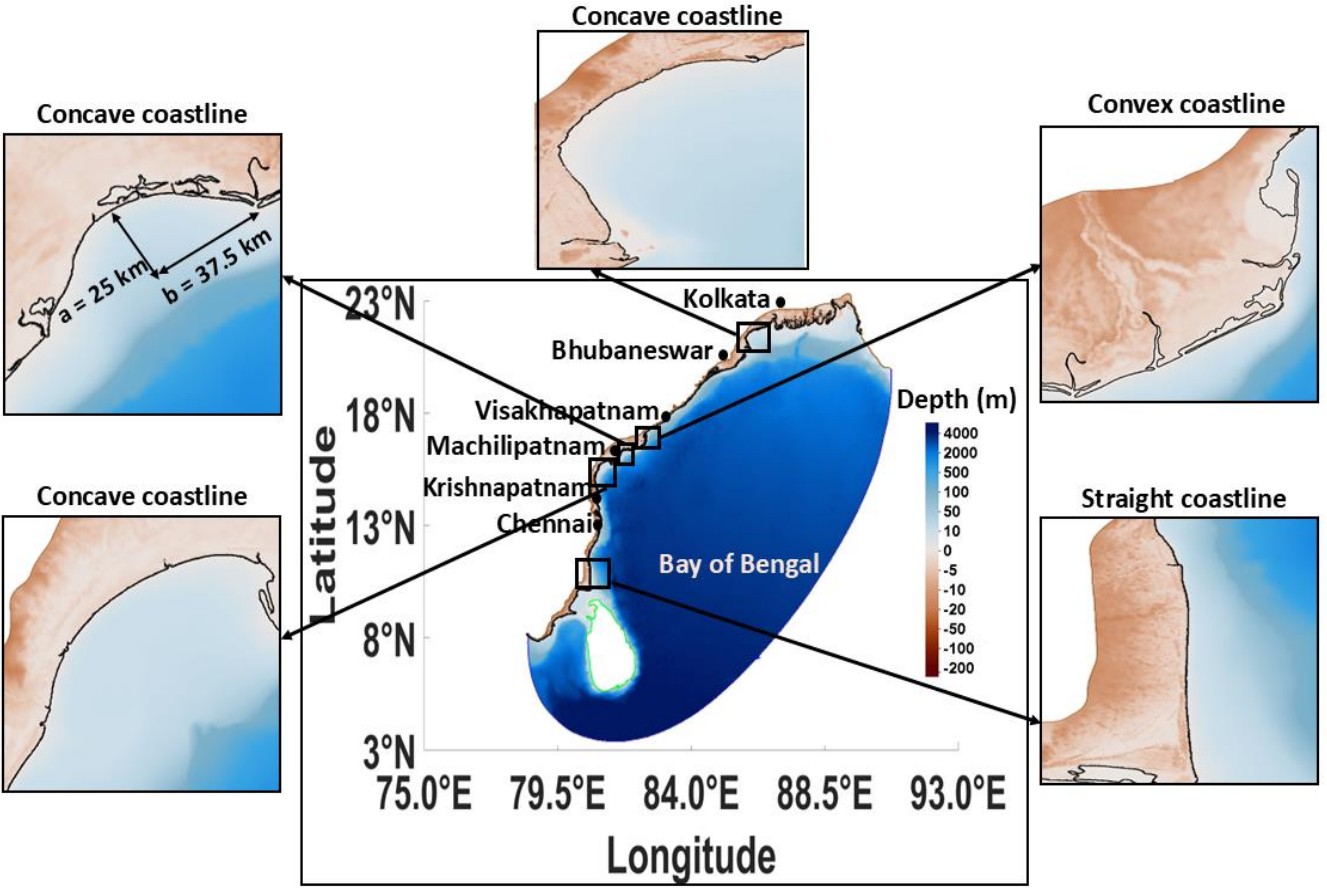

**Figure 2. Model domain along with different coastline geometry (zoomed)**

For the experiments (Exp2), different ellipse-shaped concave and convex meshes are designed with distinct **a** and **b** values, as shown in Fig. 3. Each designed shape has different **a** and **b** values than the actual one because **a** and **b** values are very small in the actual coastline. Therefore, domains of concave/convex are designed based on taking diverse **a** and **b** values. Fig. 3 (ii - vii) represent different concave shapes having **a** = 50, 100, 150, 200, 250, and 300 km, respectively, retaining the same value

of **b** = 150 km. These shapes are named CC1, CC2, CC3, CC4, CC5, and CC6 respectively. Similarly, Fig. 3 (viii-xiii) represents a convex shape with **a** = 50, 100, 150, 200, 250, and 300 km, respectively, keeping the same **b** = 150 km. These shapes are named CV1, CV2, CV3, CV4, CV5, and CV6. While Fig. 3 (xiv) shows the straight coastline domain. All the above concave and convex domains represent the ellipse shape, where **b** acts as the semi-major axis and **a** as the semi-minor axis until **a** = 150 km (circular shape), after which **a** becomes the semi-major axis and **b** as the semi-minor axis.

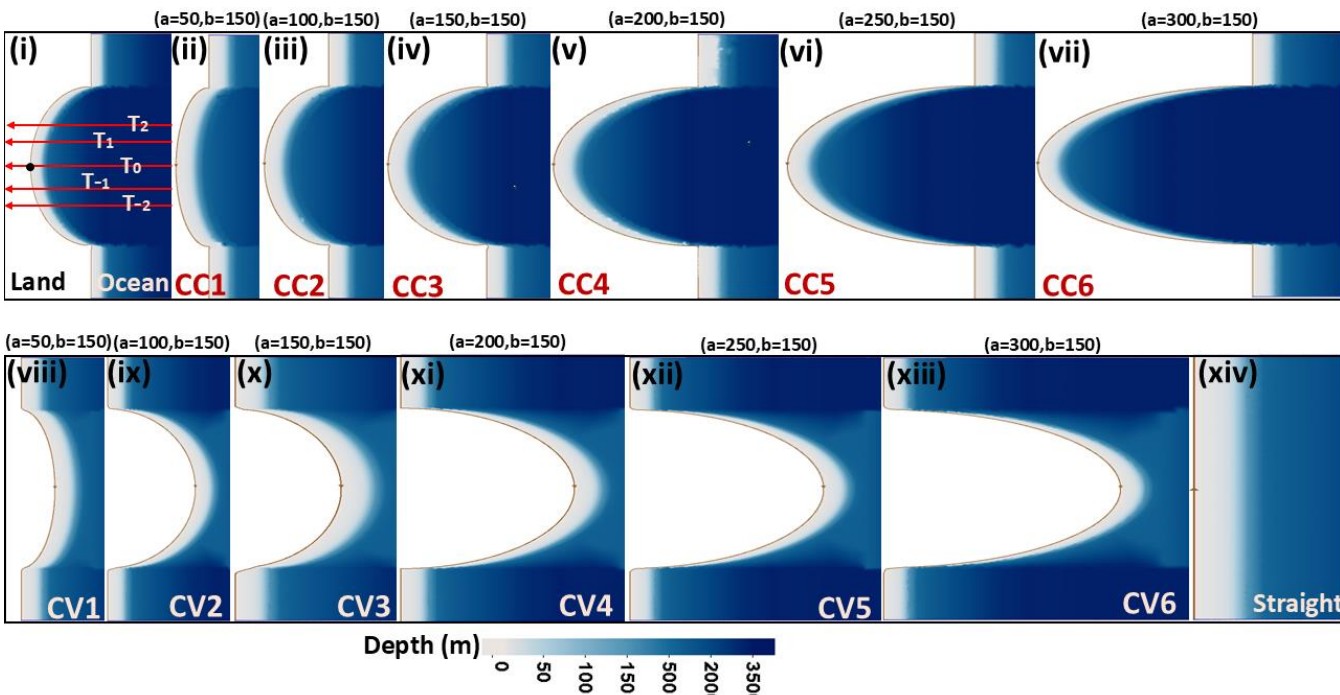

**Figure 3. (i) Parallel tracks used in the study. Black dot (●)represents the vertex of the domain. (ii-vii) Shapes of idealized coastlines considered for concave (CC1-CC6), (viii-xiii) convex (CV1-CV6), and (xiv) straight coastlines.**

A total of 13 meshes are created (6 for concave, 6 for convex, and one for straight coastline) using the Surface Water Modeling System (SMS) application (http://www.aquaveo.com/products) with a maximum resolution of 200 m near the coast and 15 km in the open ocean. The bathymetry of the domain is generated using a constant continental shelf width of about 50 km, which is an average value for the region. The depth contours follow the coastline, with a maximum depth of 3,500 m in the open ocean. As shown in Figure 3(i), five idealized parallel cyclone tracks are considered with a maximum pressure drop of 50 hPa. The translational speed of all cyclone tracks is maintained at about 13 km/hour, which is the average speed of a cyclone observed in this region from past historical data. The wind field for each track is calculated by taking a constant radius of maximum ($R_{max}$) winds as 30km, based on observation of the past cyclones in this region (Mohapatra and Sharma, 2015;

Sharma and Mohapatra, 2017). These tracks are named $T_0$ for the center, $T_1$ and $T_2$ for the tracks making landfall 30 km and 50 km north of the center, and $T_{-1}$ and $T_{-2}$ for the tracks 30 km and 50 km south of the center track, respectively. All the tracks make landfall at the same time (on the 48$^{th}$ hour). The curvature of the mesh increases from west to east for both the concave and convex-shaped coastlines, as shown in Fig. 3 (ii-xiii).

**3.2.1 Calculation of angles with respect to maximum peak surge (MPS) location based on the shape of the domain:** If the shape of the domain is represented by an ellipse, which is given by,

$$\frac{x^2}{a^2} + \frac{y^2}{b^2} = 1 \tag{1}$$

where, "**a**" is the semi-major axis and "**b**" is semi-minor axis. If **b** becomes the semi-major axis and **a** becomes the semi-minor axis, the equation remains the same. Then, the equation of a tangent angle to an ellipse is given by:

$$\tan \theta = -\frac{b^2}{a^2}\frac{x_1}{y_1} \tag{2}$$

where $(x_1, y_1)$ is any point on the ellipse above the semi-major axis as shown in Fig.4 and in this case, the angle is positive. Similarly, the corresponding angle will be negative if a point is considered below the semi-major axis.

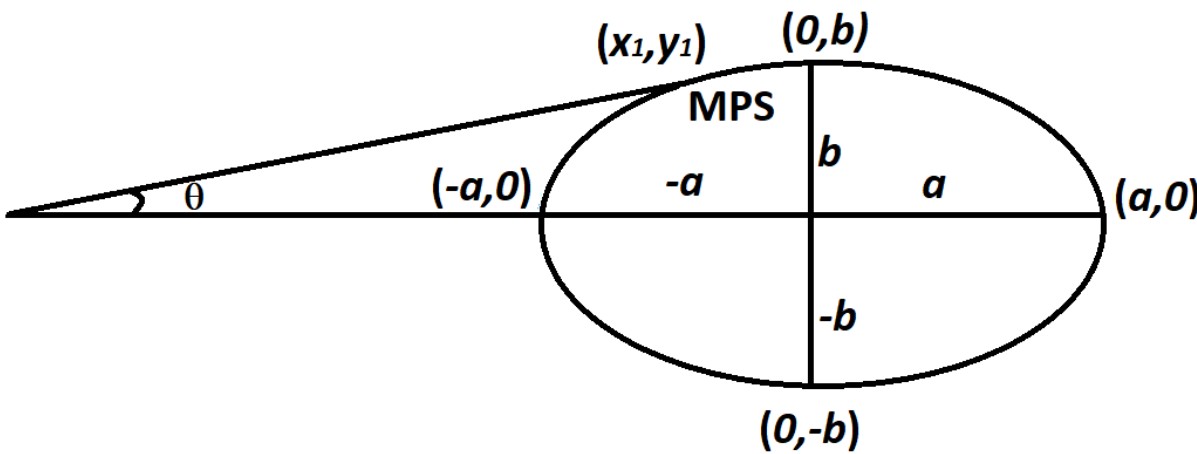

**Figure 4. Angle of a tangent (θ) to an ellipse at the point of maximum peak surge (MPS).**

**3.3 Experiments using real coastline (Exp3)**

In the last set of experiments (Exp3), the model domain incorporates actual coastline geometry and prevailing bathymetry,
which is complex in nature including coastal water features (river channels, lakes, etc.). The analysis area for this experiment
covers only from (80° E, 15° N) to (82° E, 16.50° N) along the east coast India as shown in Fig. 2. The domain mesh uses
several digital elevation models (DEM) (airborne, CARTO2, and SRTM30) and bathymetry datasets (ETOPO-2 and MIKE-
CMAP) (Mandal et al., 2020) same as in Exp1.

The model's landward boundary is considered up to 10 m contour from the shore, assuming the surge never exceeds 10 m topo.
The mesh has a high resolution near the coast of about 100 m and 18 km in the open ocean. Since the study focuses only on
the generation of surges at the coastline, the computation of associated coastal inundation is not a part of this study. Therefore,
the land is masked to counteract river openings impact or the presence of other water bodies. For the sensitivity experiments,
cyclones of parallel, oblique, and straight tracks are considered to have the same intensity as in the previous experiment. Refer
to Table 1 for the detailed set of experiments used in the study.

**Table 1.  Detail of the experiments used in the study**

| Exp1 | 1. Calculation of storm tides and validation with the observation for the cyclone Michaung. |
|---|---|
| Exp2 (Idealized coastlines) | 1. MPS generation using 5 parallel tracks and 5 different shapes of each concave and convex coastlines |
| Exp3 (real coastlines) | 1. MPS generation using parallel tracks |
| | 2. MPS generation using different approach angles. |
| | 3. MPS generation using different Rmax. |

## 4. Results and Discussion
### 4.1 Computation of storm tides for the Michaung cyclone.
In the Exp1, maximum ST are computed for the Michaung cyclone by considering tides in the model during cyclone period as
shown in Fig. 5(i). The amplitude of the local tide is coinciding with the spring tide at Krishnapatnam, which is observed about
0.5m (not shown in the figure) at the time of cyclone crossing this region. It is observed that the ST are simulated along the
coast both to the west and east side of the track as the cyclone moves northwards parallel to the southern coast before its

landfall. The ST of about 1.6m is computed on the east of the track in the concave-shaped coast near Bapatla. Moreover, ST of 1.25m are also generated along the coast near Ongole on the west of the track. The ST of 1.0m are also seen along the coast near Diviseema.

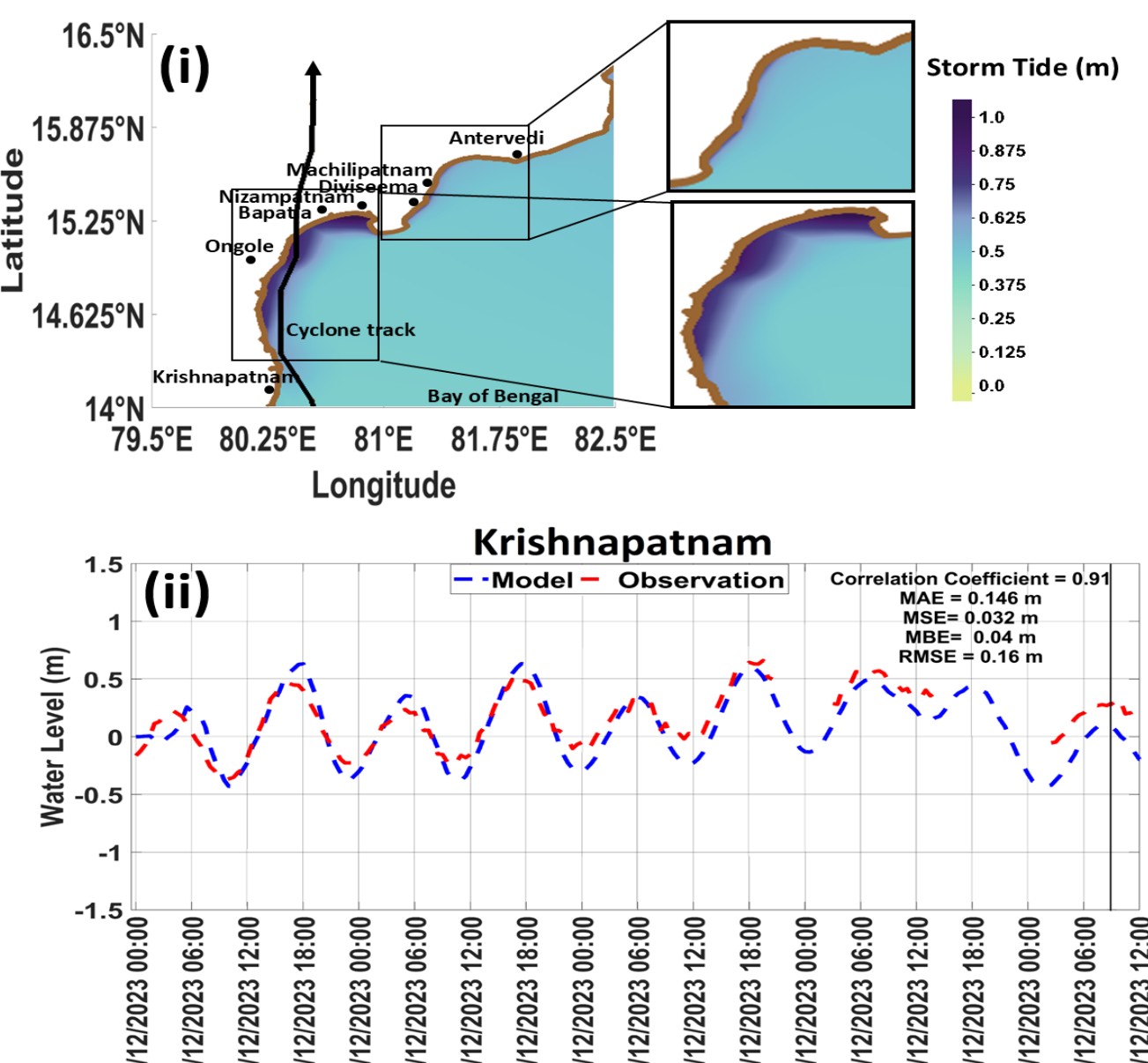

**Figure 5. (i) Model computed maximum storm tide for the Michaung cyclone. (ii) Comparison of model and tide-gauge data at Krishnapatnam. Black vertical line represents landfall time.**

Validation of the model simulated ST is made with the only available tide-gauge data at Krishnapatnam, which falls on the west of the cyclone track, located at about 30 km from the track and 160 km from the landfall point, as shown in Fig. 5 (i). As shown in Figure 5(ii), the maximum ST from the model and tide-gauge are noted as 0.63 and 0.66m at Krishnapatnam respectively, occurring 36 hours before the landfall. While the ST values are reduced at the time of landfall to 0.08 and 0.28 m respectively. However, the simulated maximum ST at Krishnapatnam is qualitatively in agreement with that of tide-gauge observation. The computed correlation coefficient, mean absolute error (MAE), mean square error (MSE), mean bias error (MBE) and root mean square error (RMSE) are 0.91, 0.146 m, 0.032 m, 0.04 m and 0.16 m respectively. Though some gaps in the tide-gauge data is seen, no missing data is found during the maximum ST or landfall time. The reason for generation of lower ST may be due to the location of Krishnapatnam, located to the west of cyclone track, which was mostly influenced by the offshore winds during cyclone period. This experiment confirms the hypothesis that the concave coasts are always subjected to more vulnerability in terms of SS whenever cyclone approaches close to it, irrespective of whether it may be to the east or west of the cyclone track.

## 4.2. Peak surge generation with parallel cyclone tracks in the idealized domains

Numerical simulations are carried out without tides using different parallel cyclone tracks as described for Exp2 to delineate its effect on the generation of SS in the concave/convex domain. The model is integrated for 3 days, out of which the cyclone makes landfall after 2.5 days. Figure 3(i) shows the computational domain along with five idealized parallel cyclone tracks making distinct landfalls within the domain. In this study, parallel tracks ($T_{-2}$ to $T_2$) are considered and the central track, $T_0$ making landfall at the vertex of the domain. The distribution of the PS, while moving from track $T_{-2}$ (south) to $T_2$ (north) within the same domain and between different shapes of the domain are depicted in Fig. 6.

The computed MPS between different concave domains is shown in Fig. 6(i). For the track $T_{-2}$, the value of MPS does not vary much for different shapes of the concave (CC1 to CC6), and the maximum difference of only 0.04 m is observed. It shows an increasing trend in MPS for the track $T_{-1}$ as the curvature of the domain increases from CC1 to CC6 with a maximum difference of 0.075 m. Whereas for the central track $T_0$, the value of PS increases up to CC4 and then decreases marginally. It is noticed that the trend is seen linearly decreasing with the increase in curvature of the domain from the track $T_{-2}$ to $T_2$.

The maximum difference in MPS between CC1 and CC6 is computed at about 0.38 and 0.60 m for the tracks $T_1$ and $T_2$, respectively. The decrease may be attributed to the curvature of the coastline, which becomes more convergent from CC1 to CC6. This indicates that as the cyclone track and the landfall position move northward from the vertex/center of the concave coast, the MPS value decreases in correspondence with the increasing convergence. Further investigation is made in this regard and is explained in the subsequent sections.

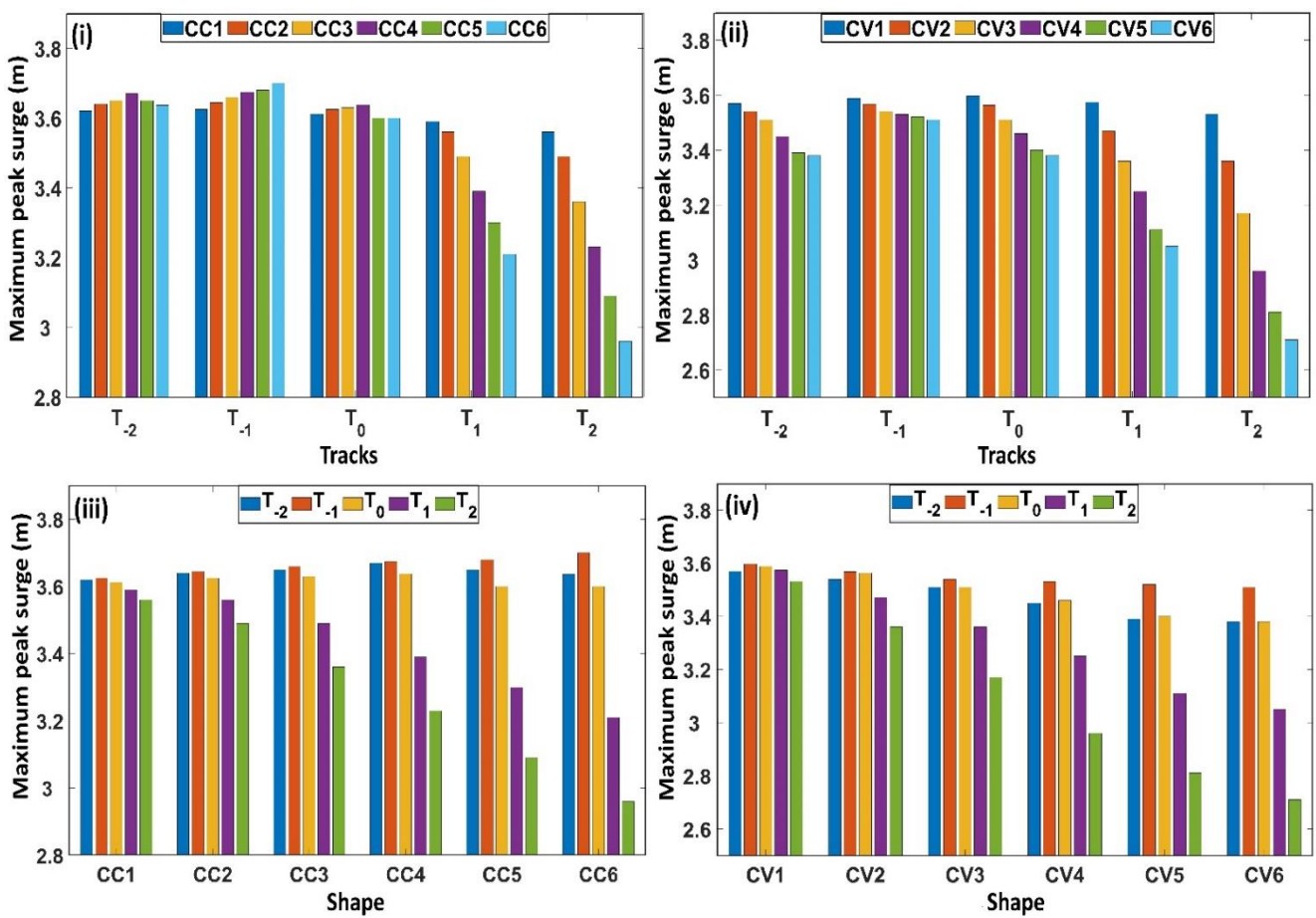

**Figure 6. Generation of maximum peak surge (MPS) for all the cyclone tracks for (i) concave and (ii) convex coastlines and due to different tracks within the (iii) concave and (iv) convex curvatures.**

Figure 6 (ii) shows the MPS trends for different convex domains. It is noted that there is a linear decrease trend in the PS from CV1 to CV6, and the decrease is substantial as the curvature increases. The maximum differences in MPS values between CV1 and CV6 range from 0.02 to 0.83 m for tracks $T_{-2}$ to $T_2$. It is observed that if the curvature of the convex shape increases, the MPS values for a particular track decrease significantly. This also underscores the influence of coastline shape and landfalling position on surge computation. Further, these simulations quantify the behavior of the MPS distribution for both concave and convex shapes.

Figures 6 (iii-iv) depict the tendency of the MPS within the concave and convex domains, respectively, as the tracks move from south to north ($T_{-2}$ to $T_2$). For all the concave/convex curvatures, the MPS increases initially from $T_{-2}$ to $T_{-1}$ followed by a decrease. However, the MPS is noticed at $T_{-1}$ in both shapes, though it is more for the concave coast, being it is a convergent

type. The value of MPS varies from 3.62 to 3.7 m for CC1 to CC6, while it ranges from 3.60 to 3.51 m for CV1 to CV6, as depicted in Fig 6 (iv). The reason for simulating higher MPS for $T_{-1}$ may be due to the value of $R_{max}$ that lies at the vertex of the curvature, as it provides maximum winds in this region. For the straight coastline (not shown here), the MPS is simulated at about 3.6m, which falls mostly between the concave and convex values. From the above analysis, it is noted that there is not much difference in the MPS computed for the CC1 as the track moves from south to north. However, the difference increases as the curvature increases for both concave and convex shapes. The maximum differences in MPS between parallel tracks ($T_{-1}$ to $T_2$) within the domain for CC1 to CC6 are 0.07, 0.14, 0.30, 0.44, 0.59, and 0.74 m, respectively. For the convex coastline domain CV1 to CV6 these values are 0.07, 0.20, 0.37, 0.57, 0.71, and 0.80 m. In general, the values of MPS increase as concave curvature increases because of the accumulation of more wave energy towards the coast. While energy gets distributed or dissipated on the convex coast with increasing curvature, which results in lower MPS along the coast.

Table 2.  **Maximum peak surge along with the tangent angle at $R_{max}$ for the different tracks for the concave shape.**

| Shape (concave) | Track | Angle (degrees) | MPS (m) |
|---|---|---|---|
| **CC1** | $T_2$ | 79.5 | 3.56 |
| | $T_1$ | 82.7 | 3.59 |
| | $T_0$ | 86.8 | 3.61 |
| | $T_{-1}$ | 90.0 | 3.63 |
| | $T_{-2}$ | -87.0 | 3.62 |
| **CC2** | $T_2$ | 70.2 | 3.49 |
| | $T_1$ | 76.2 | 3.56 |
| | $T_0$ | 84.0 | 3.63 |
| | $T_{-1}$ | 90.0 | 3.64 |
| | $T_{-2}$ | -84.4 | 3.64 |
| **CC3** | $T_2$ | 62.0 | 3.36 |
| | $T_1$ | 70.4 | 3.49 |
| | $T_0$ | 81.2 | 3.63 |
| | $T_{-1}$ | 90.0 | 3.66 |
| | $T_{-2}$ | -81.6 | 3.65 |
| **CC4** | $T_2$ | 57.3 | 3.23 |
| | $T_1$ | 64.7 | 3.39 |
| | $T_0$ | 78.7 | 3.64 |
| | $T_{-1}$ | 90.0 | 3.67 |
| | $T_{-2}$ | -79.0 | 3.67 |

| CC5 | $T_2$ | 53.2 | 3.09 |
|-----|-------|------|------|
|     | $T_1$ | 60.0 | 3.30 |
|     | $T_0$ | 76.3 | 3.60 |
|     | $T_{-1}$ | 90.0 | 3.68 |
|     | $T_{-2}$ | -76.1 | 3.65 |
| CC6 | $T_2$ | 49.2 | 2.96 |
|     | $T_1$ | 55.3 | 3.21 |
|     | $T_0$ | 74.0 | 3.60 |
|     | $T_{-1}$ | 90.0 | 3.70 |
|     | $T_{-2}$ | -74.5 | 3.64 |

**Table 3. Maximum peak surge along with the tangent angle at $R_{max}$ for the different tracks for the convex shape.**

| Shape (convex) | Track | Angle (degrees) | MPS (m) |
|----------------|-------|-----------------|---------|
| CV1 | $T_2$ | 79.5 | 3.53 |
|     | $T_1$ | 82.7 | 3.57 |
|     | $T_0$ | 86.8 | 3.59 |
|     | $T_{-1}$ | 90.0 | 3.59 |
|     | $T_{-2}$ | -87.0 | 3.57 |
| CV2 | $T_2$ | 70.2 | 3.36 |
|     | $T_1$ | 76.2 | 3.47 |
|     | $T_0$ | 84.0 | 3.56 |
|     | $T_{-1}$ | 90.0 | 3.57 |
|     | $T_{-2}$ | -84.4 | 3.54 |
| CV3 | $T_2$ | 62.0 | 3.17 |
|     | $T_1$ | 70.4 | 3.36 |
|     | $T_0$ | 81.2 | 3.51 |
|     | $T_{-1}$ | 90.0 | 3.54 |
|     | $T_{-2}$ | -81.6 | 3.51 |
| CV4 | $T_2$ | 57.3 | 2.96 |
|     | $T_1$ | 64.7 | 3.25 |
|     | $T_0$ | 78.7 | 3.46 |
|     | $T_{-1}$ | 90.0 | 3.53 |

| | | | |
|---|---|---|---|
| | $T_{-2}$ | -79.0 | 3.45 |
| CV5 | $T_2$ | 53.2 | 2.81 |
| | $T_1$ | 60.0 | 3.11 |
| | $T_0$ | 76.3 | 3.40 |
| | $T_{-1}$ | 90.0 | 3.52 |
| | $T_{-2}$ | -76.1 | 3.39 |
| CV6 | $T_2$ | 49.2 | 2.71 |
| | $T_1$ | 55.3 | 3.05 |
| | $T_0$ | 74.0 | 3.38 |
| | $T_{-1}$ | 90.0 | 3.48 |
| | $T_{-2}$ | -74.5 | 3.38 |

This analysis is further explained using a tangent angle at the location of MPS, which usually coincides with the location of maximum sustained winds along the coast based on the value of $R_{max}$. The detailed calculation of the angle corresponding to the location of the MPS for each track and shape of the domain is explained in Fig. 4. Tables 2 and 3 provide tangent angles corresponding to the location of MPS generated by various tracks against each domain. Table 2 shows that the difference in the angle between $T_{-1}$ and $T_2$ increases as the domain curvature increases from CC1 to CC6, while the angle (negative)

decreases for the track $T_{-2}$. A similar observation is also made for the convex domain, as given in Table 3.

These angles have a significant effect on the generation of the MPS, as shown in tables 2 and 3. It is noted that the maximum curvature domain (CC6) produces different MPS for different tracks ($T_{-2}$ to $T_2$) compared to that of the minimum (CC1). This infers that the curvature of the coast is playing a significant role in the generation of PS, along with the landfall location of the track in the domain. To understand further whether this angle is also responsible for the spread of the PS, the spatial distribution

of PS along the coast during the cyclone period is shown in Fig. 7 for the domains CC1, CC6, CV1, and CV6 as these two domains represent the least and the maximum curvature domains. It is noted that the least spread of PS is seen at the vertex of the domain when the angle is maximum ($\theta = 90°$) and is more as the theta decreases on either side of it. It infers that the spread along the coast increases as the curvature of the domain increases.

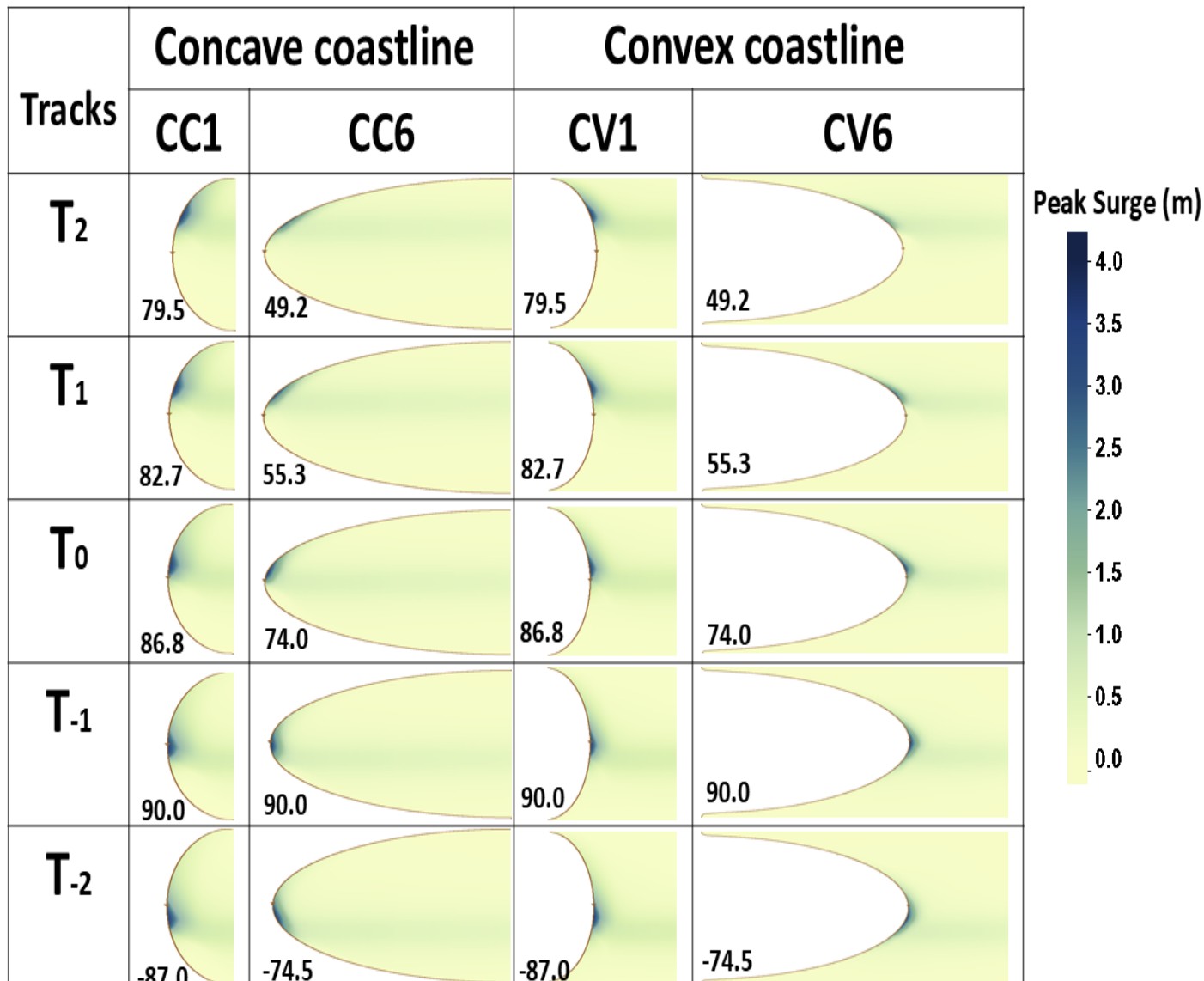

**Figure 7. Peak surge generated along the coast for two distinct shapes of concave (CC1 and CC6) and convex (CV1 and CV6) coastlines. Numbers inside the plot represent tangent angle (θ) to the location of peak surge.**

### 4.3. Onshore and alongshore winds.

As cyclone winds are the primary forcing for the generation of the SS, it is important to investigate the wind distribution along the coast with reference to each curvature of the domain. For this, wind components onshore and alongshore are estimated at

the location of the MPS and how these components vary due to different shapes of concave and convex coasts. These components provide an insight into the PS and its distribution along the coast. Storm waters are driven toward the coast by onshore winds originating at sea. Due to the accumulation of water caused by these winds along the coast, the PS enhanced. However, the alongshore wind component influences the distribution/spread and direction of water movement along the coast. Hence, the onshore wind plays a vital role in developing SS at the coast, while the alongshore wind is responsible for the spread of the surges along the coast. Figure 8 represents the temporal variation of the onshore and alongshore wind components for the domains CC1, CC6, CV1, and CV6 calculated at the location of MPS. From Figure 8, it is noted that the onshore wind, in general, increases as the cyclone approaches the coast. The negative sign represents the onshore wind towards the west (positive towards the east) and the alongshore wind towards the south (positive towards the north). At the time of landfall, the onshore winds for the CC1 are -45.1, -49.0, -48.2, -47.7, and -46.7 m/s, and the alongshore wind is -25.0, -15.0, -14.3, -15.7 and -23 m/s for the tracks $T_{-2}$, $T_{-1}$, $T_0$, $T_1$, and $T_2$, respectively. Similarly, the CC6 experiences onshore winds of -49.2, -52.3, -46.0, -40.0, and -32.4 m/s, and alongshore winds of -15.0, -4.0, -22.0, 27, and 40.4 m/s for the tracks $T_{-2}$, $T_{-1}$, $T_0$, $T_1$, and $T_2$. The onshore wind decreases and alongshore wind increases from track $T_{-2}$ to $T_2$ for CC6, which is consistent with the MPS (refer: Fig. 6), as depicted in Fig. 8. Although the maximum wind speed of 54 m/s of the cyclone is used for all the tracks, the generation of SS depends upon the strength of the onshore wind at the time of landfall, which varies with the curvature. For the convex domain (CV1), there is not much change in the onshore wind for all the tracks, which is same as CC1.

However, the change becomes significant as the curvature increases, as shown in Fig. 8(vii). At the time of landfall, the onshore winds for the CV1 are -45.0, -46.0, -44.0, -43.6, and -43.2 m/s, and the alongshore wind is -24.1, -23.2, -26.0, -26.8 and -27.3 m/s for the tracks $T_{-2}$, $T_{-1}$, $T_0$, $T_1$, and $T_2$, respectively as shown in Fig. 8 (v-vi).Similarly, the CV6 experiences onshore winds of -41.5, -44.8, -40.0, -33.8 and -30.0 m/s and alongshore winds of -29.3, -25.7, -30.5, -39.8 and -41.6 m/s for the tracks $T_{-2}$, $T_{-1}$, $T_0$, $T_1$, and $T_2$ as shown in Fig. 8(vii-viii) and Table. 4. Similarly, the onshore wind does not change much for CV1, but it makes significant changes for CV6. From all the cases discussed above, the maximum onshore wind is noticed for track $T_{-1}$ as the $R_{max}$ coincides with the vertex of the curvature. Hence, the occurrence of the MPS is observed for track $T_{-1}$. It is also noted that the distribution of the PS along the coast increases as the alongshore wind enhances, which is evident from Fig. 7.

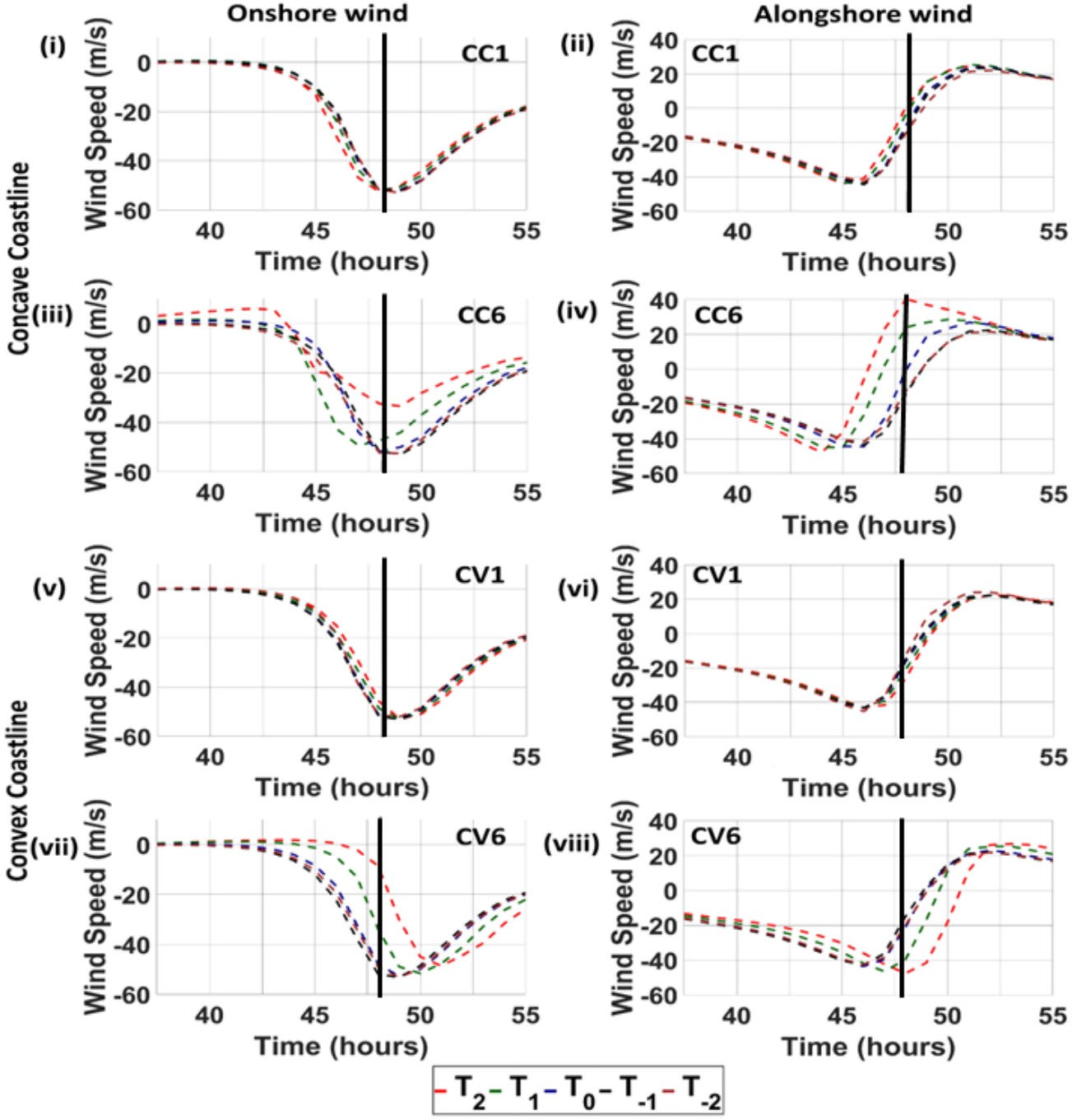

**Figure 8. Time series of onshore and alongshore cyclonic wind components for (i-ii) CC1, (iii-iv) CC6, (v-vi) CV1 and (vii-viii) CV6. Black vertical line represents the landfall time.**

**Table. 4. Onshore and alongshore wind for different tracks for the shapes CC1, CC6, CV1 and CV6.**

| Shape | Track | Onshore wind | Alongshore wind |
|-------|-------|--------------|-----------------|
| CC1 | $T_2$ | -46.7 | -23.0 |
| | $T_1$ | -47.7 | -15.7 |
| | $T_0$ | -48.2 | -14.3 |
| | $T_{-1}$ | -49.0 | -15.0 |
| | $T_{-2}$ | -45.1 | -25.0 |
| CC6 | $T_2$ | -32.4 | 40.4 |
| | $T_1$ | -40.0 | 27.0 |
| | $T_0$ | -46.0 | -22.0 |
| | $T_{-1}$ | -52.3 | -4.0 |
| | $T_{-2}$ | -49.2 | -15.0 |
| CV1 | $T_2$ | -43.2 | -27.3 |
| | $T_1$ | -43.6 | -26.8 |
| | $T_0$ | -44.0 | -26.0 |
| | $T_{-1}$ | -46.0 | -23.2 |
| | $T_{-2}$ | -45.0 | -24.1 |
| CV6 | $T_2$ | -30.0 | -41.6 |
| | $T_1$ | -33.8 | -39.8 |
| | $T_0$ | -40.0 | -30.5 |
| | $T_{-1}$ | -44.8 | -25.7 |
| | $T_{-2}$ | -41.5 | -29.3 |

**4.4 Generation of positive and negative surges along the coast.**

Figures 9(i-xx) describes generation of positive and negative surges along the coast during the cyclonic period for the shapes CC1, CC6, CV1 and CV6 for all the tracks $T_2$ to $T_{-2}$.

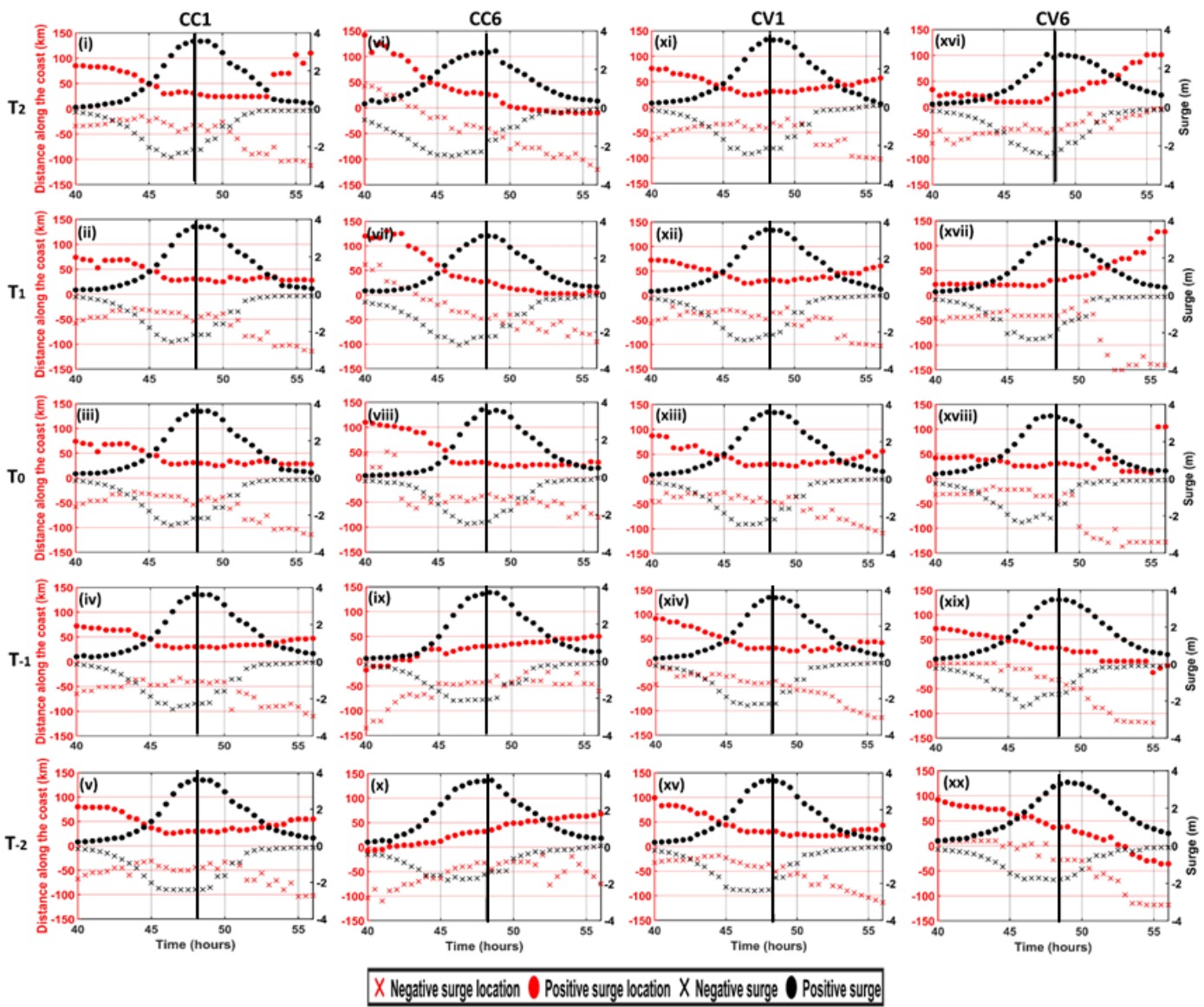

**Figure 9 (i-xx). Time of occurrence of surges along the coast with reference to the landfall location for CC1, CC6, CV1 and CV6 for all the tracks. Black vertical line represents the landfall time.**

Figures 9(i-xx) show the distance along the coast is considered with reference to the landfall location, implying positive values on the right and negative on the left along with the surge values. The left y-axis (red), zero indicates the landfall location of the cyclone track and right y-axis depicts the surge value. Figures 9(i-v) describes surge development for CC1 for all tracks. It is observed that the positive surge occurs close to the landfall on the right side, while the negative surge on the left generates about two hours prior to the landfall time. Similar observation is made from Fig. 9(vi-x) for CC6. The simulations for different

concave coasts (CC1, CC6) suggest that the positive surge decreases for the tracks from $T_{-2}$ to $T_2$, while the negative surge increases. This can be attributed to reduction of available domain on the east side of any track in CC6 to drag water masses by the onshore winds towards the coast as we consider from $T_{-2}$ to $T_2$.

As seen from all the simulations for CC6, it is observed that the maximum surge along the coast is always propagating towards the center/vertex of the curvature. In the case of convex coasts (CV1, CV6), the surges generated along the coast, in general,

diverge/spread from the vertex of the curvature. As the curvature increases from CV1 to CV6, a reduction of PS is seen for all the tracks because of lower onshore and higher along shore winds (refer Fig. 8(v-viii)). The onshore winds are responsible for the accumulation of surge waters towards the concave coasts, while the alongshore winds for divergence/spreading of the waters along the convex coasts.

**4.5 Computation of energy density per unit length of a surge wave in the concave coast**

As observed from the earlier simulations, the value of PS in the concave domain increases as the storm SS waves are propagated towards the coast. The increase in the SS can be related to an increase in the corresponding energy flux in the domain. If the shape of the domain becomes narrower and shallower away from the sea, the surge will increase up in the domain. This may be known as the funnelling effect. If we neglect frictional losses, the wave energy flux will theoretically remain constant as

the wave progresses up in the domain, and the wave energy density is given by:

$$E = \frac{\rho g A^2}{2} \tag{3}$$

where, $A$ is the amplitude of the surge wave; $\rho$ is the density of the water and $g$ is the acceleration due to gravity. The energy density per unit length of the domain is given by ($EB$), where $B$ is the width of the domain. Then the energy flux in the domain given by ($EBc$) will be constant (c being the phase speed or energy propagation speed in the shallow water), i.e.,

$\frac{(\rho g A^2) B (gh)^{\wedge \frac{1}{2}}}{2} = $ constant   or   $A \alpha\ B^{-1/2}\ h^{-1/4,}$ where $h$ is the depth $\tag{4}$

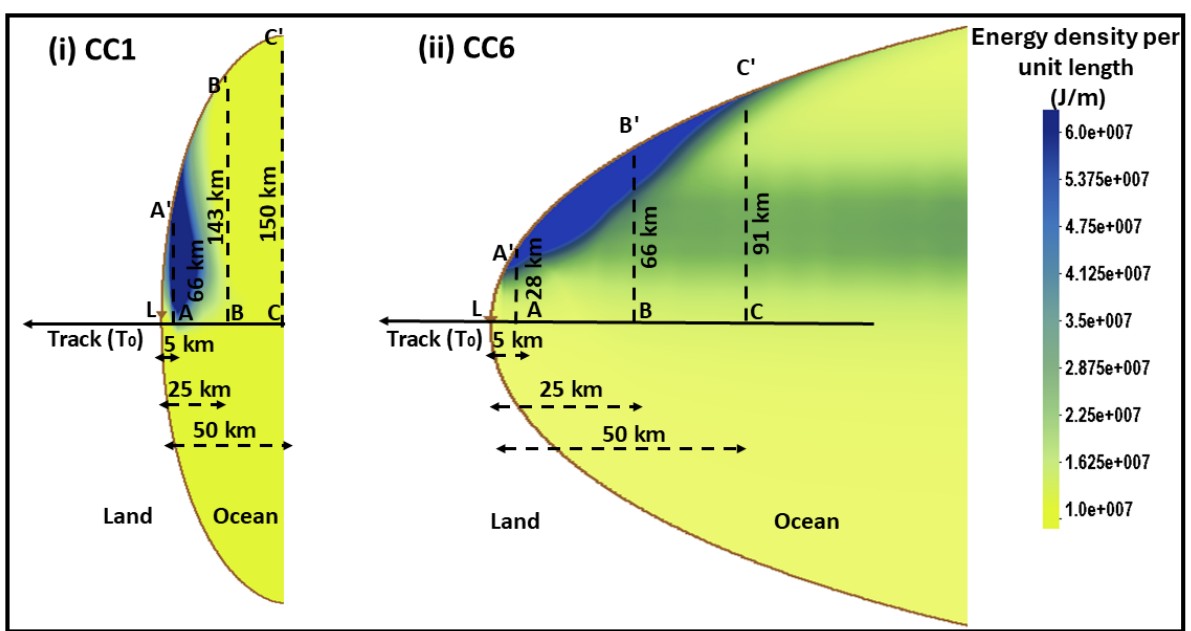

**Figure 10. Energy propagation associated with the peak surge for (i) CC1 and (ii) CC6 as the cyclone proceeds towards the coast. The L refers to the cyclone landfall.**

Energy density per unit length (*EB*) is computed for concave coasts of CC1 and CC6 as shown in Fig. 10(i-ii) using the computed PS (*A*) at each grid and multiplied by the grid resolution (*B*) for the track $T_0$. It is observed that the maximum *EB* is computed near the coast of 60 MJ/m and 58 MJ/m for CC1 and CC6 respectively at the time of MPS. As seen in the figure, the integrated value of *EB* is computed across the lines AA', BB' and CC' falling at 5km (LA), 25km (LB) and 50 km (LC) from the vertex (L) of the domain for the shapes CC1 and CC6. These values are 1953, 197 and 56 MJ/m for CC1, while it is

983, 300 and 85 MJ/m for CC6 respectively shown in Table. 5. The higher energies (AA') are seen close to the landfall for CC1, while it is decreased significantly for CC6. However, higher values are seen in CC6 much away from the landfall (BB', CC') compared to that of CC1, implying more spread of *EB* for CC6. For track $T_0$, the funnelling effect is not seen in terms of surge generation (refer to Table 5). The peak *EB* is associated with the radius of maximum winds (Rmax) in either case. However, the spread/extent of the peak *EB* is seen more in CC6 compared to CC1 due to funnelling effect.

**Table 5. Maximum peak surge along with the tangent angle at $R_{max}$ for the different tracks for the convex shape.**

| CC1 | Distance from the vertex L (km) | Energy density per unit length (MJ/m) |
|---|---|---|
| | OA (5) | 1953 |

| | OB (25) | 197 |
|---|---|---|
| | OC (50) | 56 |
| CC6 | OA (5) | 983 |
| | OB (25) | 300 |
| | OC (50) | 85 |

The value of width (*B*) reduces as the cyclone gets closer to the concave shore resulting in an increase in the energy density and *EB*. This increase of energy is directly responsible for the enhancement of surges in the concave-shaped domains. Since *B* has a more significant influence on *A* than *h*, we have considered only *B* in the above computations, leaving out *h*. This explanation holds good for all other concave shapes as well. This formula may not be applicable directly for convex domains. However, it is expected that more energy will dissipate along the coast for the convex domains instead of accumulation as in the case of concave. The simulations for all the concave domains (not shown), in general, suggest that the energy is more focused towards the coast as the tracks move from $T_2$ to $T_{-2}$.

In the next experiment (Exp3), complex coastal stretch of having concave and convex geometry is selected along the east coast of India to investigate the impact of PS in the region. This experiment consists of three parts. The first part computes PS using three independent parallel tracks (Fig. 11). The second part deals with the tracks approaching at different angles at different locations in the domain (Fig. 12). The final part examines the generation of the PS with three straight tracks parallel to the coast with different $R_{max}$ values (Fig. 13). In all these experiments, the strength of the cyclone remains the same.

**4.6 Computation of peak surges by parallel tracks using actual coastal stretch along the east coast of India.**

As shown in Figure 11, three parallel tracks, 1, 2, and 3, are considered in the concave-shaped coast, which makes landfall at different locations on the south, center, and north sides of the concave coast, respectively. $R_{max}$ is considered 30km for all the three tracks. Track 1 is making its landfall near Diviseema, with the MPS of 3.94 m occurring in the middle of the concave coast, as shown in Fig. 11(i). The reason for this is attributed to the prevailing maximum onshore winds associated with track1, which is found close to the vertex of the concave coast as the $R_{max}$ is 30km. Tracks 2 and 3 make landfall near Machilipatnam and Antervedi with MPS of 3.51 m (Fig. 11(ii)) and 3.13 m (Fig. 11(iii)), respectively.

As the tracks move from south to north, a reduction in the PS/MPS, and shifting of its location is perceived. The reason for the reduction is due to the change of landfall location on the concave coast which leads to a reduction in the onshore winds as the

tracks change from 1 to 3. Hence, accumulation of the storm waters reduces as the tracks move to the north, deducing the effect of the concave coast reduces as the landfall location of the track moves to the north of the coast. It is also important to note that the MPS is also generated to the west of track1 and 2 near Nizampatnam due to the presence of another concave coast close to Bapatla. A convex-shaped coastline is also enclosed between the concave coasts of Nizampatnam and Diviseema.

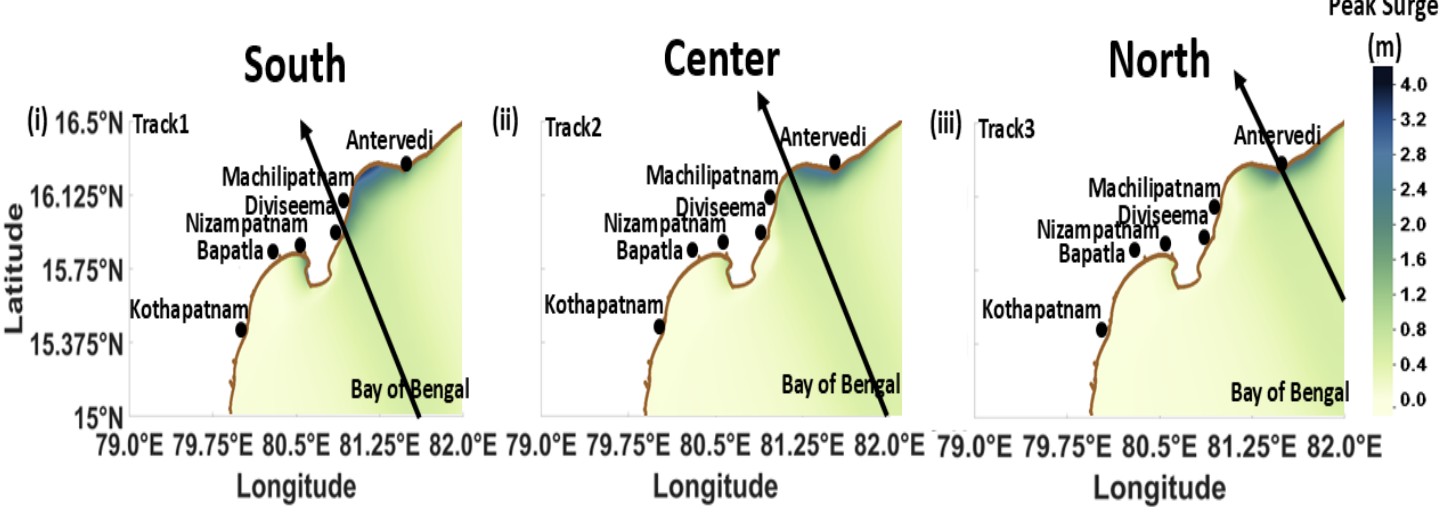

**Figure 11. Occurrence of peak surge along the actual concave coastline with the different parallel cyclonic tracks landfalling at (i) west (Track 1), (ii) center (Track 2) and (iii) east (Track 3) of the concave coastline.**

However, the PS reduces in this region as the tracks shift to the north. The experiments infer that the concave coastline geometry is more susceptible to higher SS especially when the track makes landfall on the south of the coast. Additionally, SS computed along the concave-shaped coastline is higher compared to that of the convex coastline, which aligns with the observations made with the idealized concave and convex domains (refer to section 5.2).

**4.7 Computation of peak surges by the tracks with different approach angles using actual coastal stretch along the east coast of India.**

In this part of Exp3, the impact of various approach angles and the location of the landfall on the generation of PS in the concave coastline is explored. These approach angles and landfall locations considered in this study are based on the history of the cyclones formed in this region, which include the 1977 Andhra Pradesh cyclone, Ogni (2006), Khaimuk (2008), Laila (2010), Helen (2013), Lehar (2013), and Roanu (2016) cyclones (IMD Report).

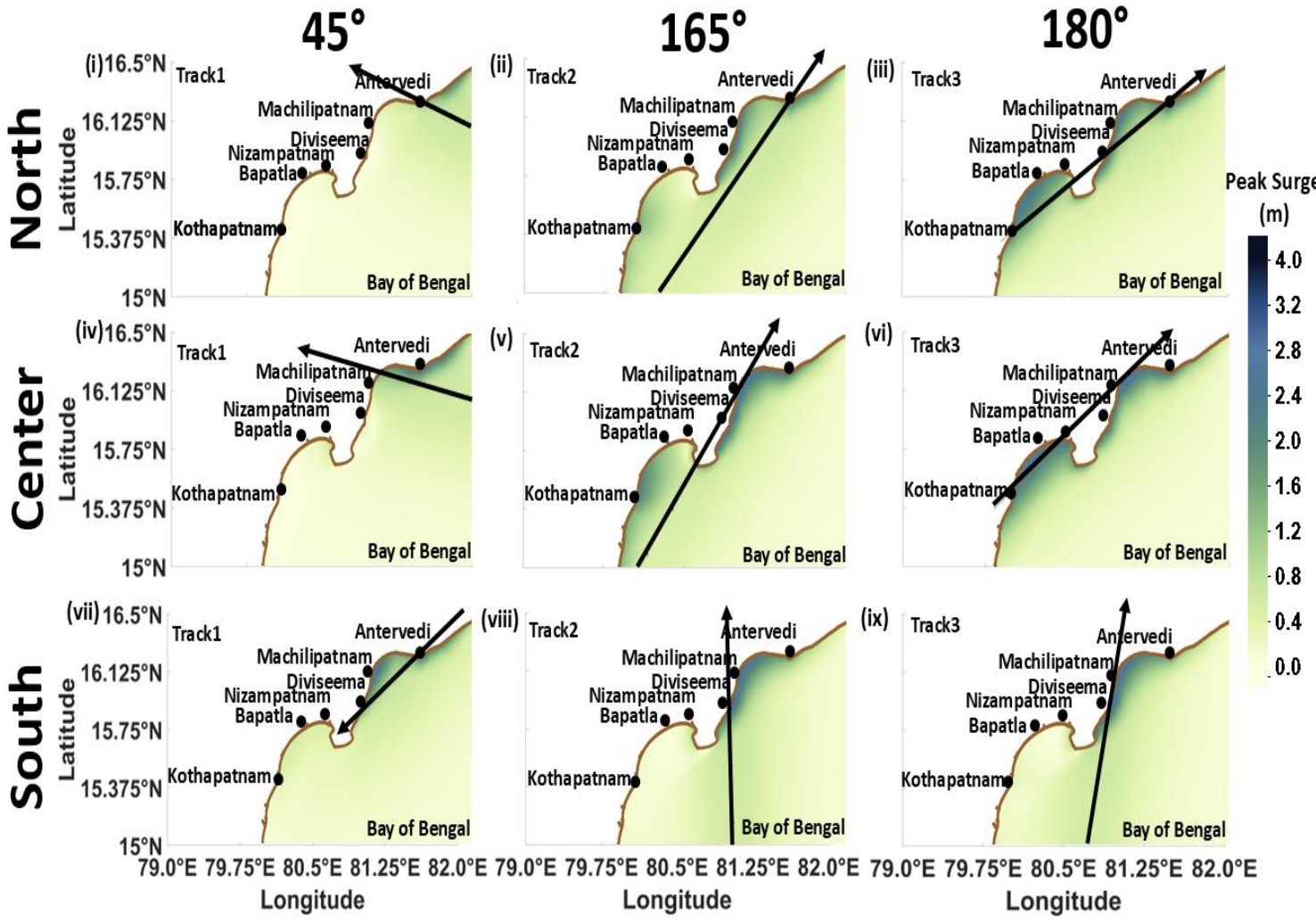

**Figure 12. Effect of approach angle (45 (track 1),165 (track 2) and 180 degrees (track 3)) on peak surges due to tropical cyclone landfalling at (i-iii) west, (iv-vi) center and (vii-ix) east of the concave coastline.**

The model domain is the same as that of section 4.6. Here, three landfall locations on the concave coast are considered with three different approach angles. Though some tracks cross the coast multiple times, the PS is studied at selected landfall locations in the concave domain, as discussed in the above section. As a part of this exercise, the PS is simulated in the concave-shaped coastline with approach angles of 45, 165, and 180 degrees made at the landfall location (east, center, and west) represented as tracks 1, 2, and 3 as shown in Fig. 12. The reason for selecting these tracks is that some generate PS over the region that is west of the track. The track angle is measured clockwise, with the tangent drawn parallel to the landfall location. These tracks are chosen so that they have more influence on the occurrence of PS in the concave domain. The tracks striking

on the east side of a concave coastline near Antervedi generate MPS of about 2.56, 3.08, and 3.15 m for tracks 1, 2, and 3, respectively, as shown in Fig. 12 (i-iii).

The MPS is also computed in the neighboring concave-shaped coastline near Bapatla at about 2.01 and 3.02 m for tracks 2 and 3, respectively. The MPS for track 1 is computed on the east side, while for tracks 2 and 3 it is on the west side due to the concave nature of the coastline. In Figures 12(iv-vi), tracks 1, 2, and 3 are making landfall at the center of the concave domain near Machilipatnam, and associated MPS are about 2.83, 3.59, and 3.62m, respectively. Tracks 2 and 3 also generate MPS in the adjacent concave-shaped coastline near Bapatla of about 2.75 and 3.27 m, respectively. The simulations shown in Figures 12(vii-ix) correspond to the landfall location near Diviseema with different approach angles. The simulated MPS for tracks 1, 2, and 3 are approximately 3.34, 3.61, and 3.67 m respectively, which exhibit higher SS in the concave domain compared to that of any other landfall locations. This suggests that the occurrence of MPS is not always confined only to the east side of the cyclone track if a cyclone makes landfall towards the concave geometry with unique approach angles. These simulations infer that the generation of PS depends on both track approach angle and the landfall location in the concave domain.

## 4.8 Computation of peak surges by parallel tracks with different $R_{max}$ using actual coastal stretch along the east coast of India

In the past, many parallel cyclonic tracks along the east coast of India were seen that passed through the coastal stretch covering from Kothapatnam to Antervedi, which has a complex geometry with concave/convex-shaped coastlines. Though the cyclones may not have landfall in this region, this coast is prone to higher SS due to its irregular coastline geometry. Hence, this experiment is designed to study the effect of PS with equally spaced (20km) tracks moving parallel to this coast with different $R_{max}$ values of 20, 30, and 40km for tracks 1,2 and 3, respectively, as shown in Fig. 13. The PS in the domain is depicted in Fig. 13 (i-iii) as the cyclonic tracks are moving parallel to the coast with $R_{max}$ 20km. It is important to note from the simulations that all these parallel tracks generate SS along the coast. However, the MPS of 2.85m is simulated with track 1 and the minimum of 2.0m by track 3. Similarly, the experiments are carried out with $R_{max}$ 30km and 40km, as shown in Fig. 13 (iv-vi) and (vii-ix), respectively. The pattern of surge development at the coast is very similar to that of $R_{max}$ 20km. However, the MPS simulated for track 1 is 3.1m and 3.2 m for $R_{max}$ 30km and 40km, respectively. It is also observed that the MPS for track 3 are 2.5m and 3.0m for $R_{max}$ 30km and 40km respectively. It is worth noting that the PS is generated only on the west of the concave coasts due to strong associated onshore winds, and the maximum is produced along the coast with $R_{max}$ 40km. it is observed from the simulations made for $R_{max}$ more than 40km (not shown here) that the effect of SS on the coast is reduced as the tracks move away from the coast. Moreover, the concave coast in the north covering Diviseema to Antervedi experiences higher SS compared to that of adjacent concave coast lying in the southern part of the domain at Bapatla. The increase of Rmax

does not affect the magnitude of the wind but modifies the cyclonic horizontal wind distribution. Higher surges are expected

generated at far off coastal places from the landfall point as the Rmax increases.

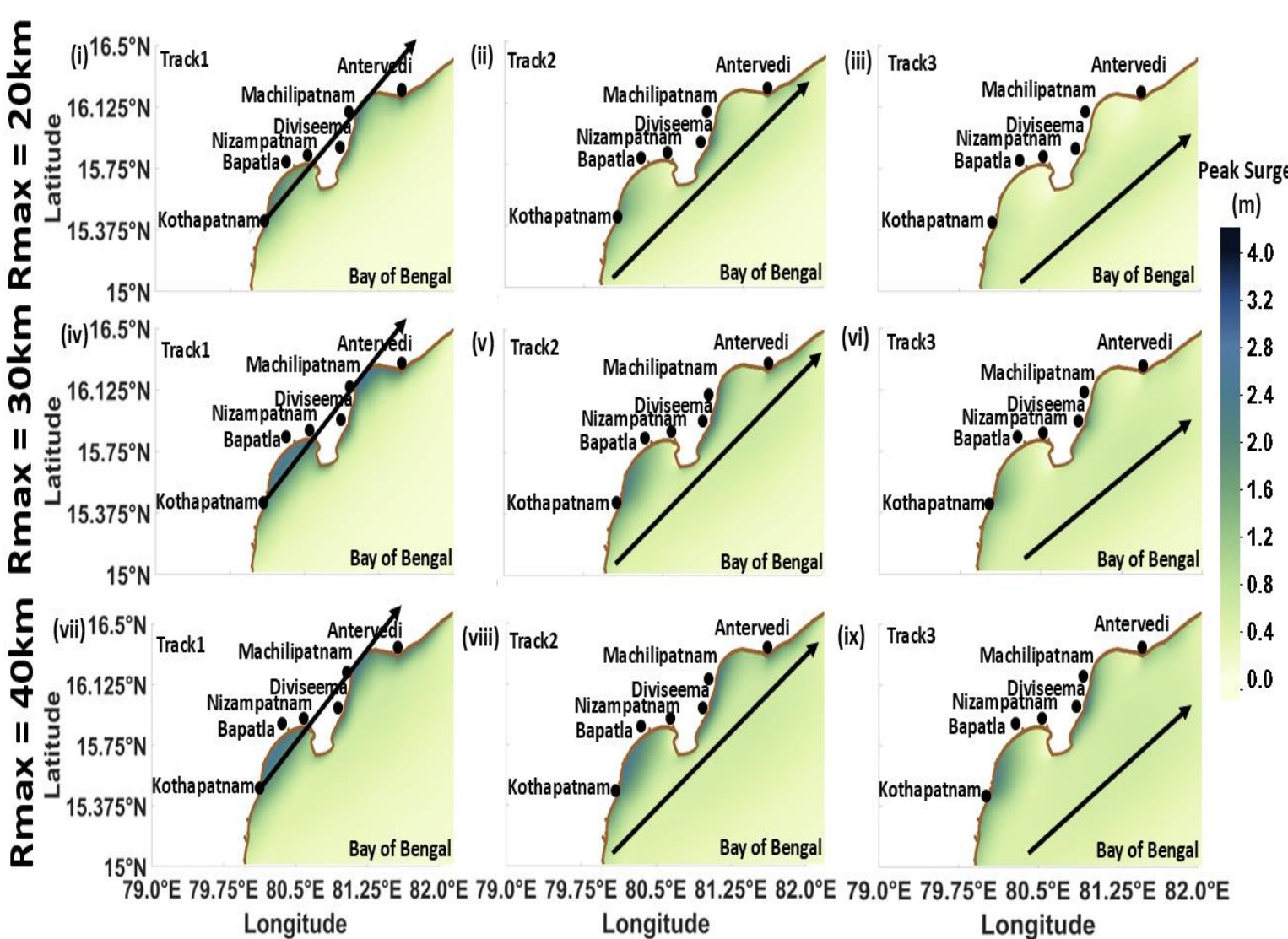

**Figure 13. The occurrence of peak surge over concave coastline with different parallel tracks and radius of maximum wind (Rmax) of (i-iii) 20km, (iv-vi) 30km and (vii-ix) 40km.**

## 5. Conclusions

The study emphasizes the complex interrelationship between the growth of SS and irregular coastline geometry like concave and convex. For this, a numerical model, ADCIRC, is used for the generation of SS with differently shaped shorelines along with diverse cyclonic tracks with the same intensity. The study is confined to the computation of PS during the cyclone period and delineates the distribution of PS along the coast. The present study is divided into three experiments by considering idealized/actual coastal domains of concave/convex. Exp1 is performed to compute ST associated with the recent cyclone, Michaung. The simulations of ST are computed by considering the tides in the model with a maximum of 1.65m near Bapatla (right to the track) and 1.25m near Ongole (west to the track), which are consistent with the reports of the India Meteorological Department (RSMC report, 2023). A comparison of model ST with the tide-gauge data available at Krishnapatnam suggests that the model can simulate the ST reasonably well even in the complex coastal stretch, as considered in the study. In Exp2, idealized different-shaped concave/convex domains are considered with many parallel tracks. The curvature effect of the concave coast on the generation of MPS is observed more for the tracks moving from south to north ($T_{-1}$ to $T_2$) in the domain. However, MPS associated with southern tracks ($T_{-2}$ to $T_0$) produce higher MPS compared to that of $T_1$ and $T_2$ because of more accumulation of storm waters in the domain. The MPS associated with $T_{-1}$ is the highest among all the tracks, as the $R_{max}$ value coincides with the domain's vertex. In general, higher curvature of the domain (CC6) produces larger amplitudes of the PS, as seen with the track $T_{-1}$. Similar experiments with the convex-shaped domain reveal a monotonic decrease of the MPS as the curvature of the domain increases from CV1 to CV6 with the highest difference for track $T_2$. It is noticed that the MPS decreases from $T_{-1}$ to $T_2$ as higher dissipation of energy is expected along the convex domain as the tracks move from $T_{-1}$ to $T_2$.

As the curvature effect plays a significant role in the PS generation along the coast, the tangent angle with respect to all the tracks is computed for the domains having the least (CC1/CV1) and the highest (CC6/CV6) curvature, which can play as a proxy for generation of MPS by any parallel track. If the angle is close to 90 degrees as in the case of $T_{-1}$, the computed MPS is seen higher, and it decreases on either side as the angle decreases. With the decrease of angle, the spread of the PS along the coast is also seen more, which is explained by the alongshore wind. At the same time, the MPS along the coast is attributed to the onshore wind. It is important to observe that the onshore wind is the maximum for $T_{-1}$ and decreases as it moves to $T_2$ with increasing curvature and hence the MPS decreases. While the alongshore wind increases from $T_{-1}$ to $T_2$, resulting in higher spread of the PS along the coast. Time evolution of SS for concave coasts suggest that both positive and negative surges incline towards the vertex of the domain as the track moves from $T_2$ to $T_{-2}$ and away from the vertex for convex coast. Computation of *EB* for the concave coasts demonstrates the funneling effect and suggests that the spread of the energy along the coast enhances as the curvature of the coast increases from CC1 to CC6. These idealized experiments signify the effect of important parameters like tangent angle, wind components, and shape of the curvature, which are closely related to the growth and spread of the PS along the coast for any track.

Exp3 is carried out with the real coastline of having complex geometry covering from Kothapatnam to Antervedi, which is one of the most prominent cyclone prone regions along the east coast of India. The simulations are designed using parallel and oblique tracks landfalling at different locations in the region. Initially, Exp3 considers three parallel tracks to evaluate the PS development in the domain of the concave coast. The study reveals that MPS are simulated if the track is close to the south of the concave coast and decreases as it moves northwards, which is consistent with Exp2. The second part of Exp3 considers the exact landfall locations as in the first part with oblique tracks along with its different approach angles of 45, 165 and 180 degrees. These experiments suggest that the PS along the coast enhances as the approach angle increases, inferring that the track parallel to the coast generates more PS, not only to the east but also to the west of the track for certain tracks. In the final part of the Exp3, three parallel cyclone tracks are considered to have different $R_{max}$ values of 20, 30, and 40km as these tracks produce a larger amplitude of the PS along the coast. The study reveals that the cyclone with higher $R_{max}$ generates more PS to the west of the track, particularly on the concave coast, signifying the vulnerability of the region though the cyclone moves parallel to the coast without making its landfall. As there are many cyclones in the past landfalling in the same region.

Our experiments suggest that the concave coastlines are responsible for enhanced surges due to the accumulation of waters as a result of funnelling effect, which is consistent with the earlier studies (Dassallas and Lee, 2019; Eurotop, 2018; Sebastian et al.,2019). Our study demonstrates less surge generation along the convex and straight coastlines compared to that of concave, which is also in agreement with the studies of Subramaniam et al. (2019), Pandey and Rao (2019), and Ueno (1981). Our study includes the effect of parallel and oblique tracks on surge generation due to different idealized concave and convex coastlines along with actual coastlines. This provides detailed insights regarding the mechanisms of surge generation. The funneling effect in terms of *EB* in concave-shaped domains is demonstrated. Our experiments suggest that west side of the cyclone track experiences higher surges for some approach angles, which is not reported earlier.

Though the study is constrained to idealized concave/convex domains with idealized bathymetry used to investigate the effects of curvature on the generation of SS, it may not fully capture the intricacies of actual coastal geometries. The absence of tides in the experiments will not influence the understanding of the storm surge generation, however it will impact on its amplitude, which is dependent on the phase of the tide. Real-world coasts may have irregular coastline with many inlets, and complex bathymetry that might influence SS behaviours differently. Further, the study assumes uniform cyclone intensity and a limited range of Rmax values (20-40 km). Our study is confined to the generation of surges, but it neglects the influence of tides and interactions like tidal asymmetry, and spring-neap tide interactions. The study should be extended in future by including the tides and wind waves with different cyclone intensity for better understanding the generation of extreme water levels in the region of interest.

This study helps to unearth the complex relation in generating SS with respect to the coastline geometry, approach angle and $R_{max}$ of the cyclone. This comprehensive study applies to any part of the coastal region having complex geometry. This study

may also be important as concave/ convex coastline domains may face shape alteration due to several factors, including erosion, sediment deposition, sea level rise, and human activities. In addition, the investigations may assist in designing better preparedness and response strategies that target coastal communities. Such knowledge can thus be utilized in developing more accurate predictive and warning systems ahead of time.

## Author contributions

**Pawan Tiwari:** Conceptualization, Formal analysis, Investigation, Methodology, Software, Validation, Visualization, Writing – original draft, Writing – review and editing. **Ambarukhana D. Rao:** Conceptualization, Investigation, Methodology, Supervision, Visualization, Writing – review and editing. **Smita Pandey:** Writing – review and editing. **Vimlesh Pant:** Writing – review and editing.

## Acknowledgments

The authors are thankful to the Indian National Center for Ocean Information Service (INCOIS) for providing tide-gauge data, the India Meteorological Department (IMD) for providing the best track data. The authors also thank IIT Delhi HPC facility for affording computational resources.

## Data availability statement

The best track data used in the study can be obtained from the IMD website (https://rsmcnewdelhi.imd.gov.in/rsmc-tropical-cyclones.php) and the tide gauze data used in this study can be obtained from the INCOIS website (https://incois.gov.in/portal/datainfo/drform.jsp).

## Competing Interests:

The contact author has declared that none of the authors has any competing interests.

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
