# Peer review of "Investigation of complex coastline geometry impact on the evolution of storm surges along the east coast of India: A sensitivity study using a numerical model"

_EGUsphere, 2024_

## Referee Comment (RC2)

Review of the manuscript "Investigation of complex coastline geometry impact on the evolution of storm surges along the east coast of India: A sensitivity study using a numerical model" by Tiwari et al., 2024.

**Overview:**

This paper discusses the effects of coastline shape and approach angle of a storm on peak storm surge, first using an idealised model, then expanding the analysis to the context of the coastline of India, and finally validates the model using Storm Michaung, a recent cyclone in the region. Previous literature on this topic analyses the effects of concave and convex coastlines on storm surge, while other papers in the field analyse the approach angle of the storm on the peak storm surge. This manuscript takes these analyses further by also considering the sharpness of the curvature of the coastline and combining this with varying the approach angle.

In general, this manuscript addresses some interesting questions regarding the effects of coastal geometry on storm surges, using both idealised and "real world" models, and presents some novel results along with results that corroborate with those in the aforementioned literature. It should be improved in its discussion of the results, with a more in-depth analysis in the text highlighting how the results fit into the context of the field. The presentation of some of the results also needs improvement.

Overall, I recommend that this manuscript undergoes major revisions regarding the text in the results and discussion sections. Although the results are generally satisfactory and I do not suggest new analysis, the writing needs some significant improvements. I hope this is not disheartening since I believe that there are some really interesting results in the paper which are worthy of publication, but I feel that the work needed on the text in some sections will likely require more time than the deadline for minor revisions would allow.

**General comments:**

In the introduction you refer to several papers which assess the impact of coastal geometry on storm surges, but these papers are not mentioned again after the initial literature review. You should refer back to these papers in your results and conclusion, to discuss how your results compare to theirs, whether or not they are consistent with them and why, etc. It would also be interesting to understand better from the introduction how your study is unique from the others discussed. I don't understand well from reading the manuscript which of your results are original and if some of your results mostly replicate results from other studies, or how your analysis fits into the wider context of extreme sea level events in the region. This is an important context that is missing.

In the results sections, instead of referring to the left/right of the storm, use west/east. Moreover, many results are listed in the text rather than presented in the form of tables or figures, which

makes them much more difficult to parse. Results that contain lists of more than a couple of values should be presented in tables or figures.

A general remark for the figures is to consider different colour map choices for contour plots rather than the rainbow/'jet' map chosen in several figures. Please check the Ocean Science figure guidelines for more information on choosing the best colour maps: https://www.ocean-science.net/submission.html#figurestables.

In the figures you use a black dot in several timeseries to represent the landfall time. A vertical line would be much clearer. This is the case for figures 5, 6, and 11b.

**Specific comments:**

**Introduction**
The first paragraph of the introduction doesn't contain any references to literature. Perhaps you can refer to a book chapter or literature review that gives a more comprehensive overview of the topic of storm surges and coastline geometry. I see that you want to introduce the topic and highlight its importance, and I think this paragraph otherwise achieves this goal well, it just needs to cite the correct sources.

Line 76: add a reference to your existing Figure 1 when you describe the coastline of India.

Line 77, "*Several cyclones in the past hit these coastal stretches with different approach angles. Some of the cyclones move parallel to these areas without making any landfall.*": please provide some examples of cyclones which have followed these paths, or references from literature to studies on these types of storms.

**Synoptic History of the Michaung Cyclone**
I think that this section should be added to the introduction as a subsection; it's a very short paragraph to have a section to itself, and it is part of the general overview of the topic.

I recommend including a figure of Cyclone Michaung's entire track to accompany this description, labelling the significant dates and locations mentioned in the text.

**Model Description**
Line 121, "*The sea level rises one cm for each hPa decrease in atmospheric pressure. The wind load on the sea surface leads to a notable fluctuation in sea level, influenced by the maximum wind speed and geographic distribution of winds*": please include an equation or a reference for these statements.

**Data and Methodology**
It would be useful to split this section into three subsections, one for the idealised experiments, the second for the real coastline experiments and a third for Cyclone Michaung. I would also

find it useful to see a table summarising each set of experiments for reference, particularly the idealised experiments since there are lots of them.

Line 133, "*The bathymetry of the region also follows a concave-shaped coastline.*": could you clarify the wording of this sentence? I'm not sure whether you mean that the coastline is concave, or that in addition, the bathymetry is smooth and follows the same shape.

Line 135: **a** and **b** are not defined in the manuscript. Please include some equations or a schematic figure to define them. I would suggest moving Figure S1 to the main manuscript since it shows **a** and **b** as well as being referred to later in your text.

Line 146, "*The bathymetry of the domain is generated using a constant continental shelf width of about 50 km*": is it 50km exactly or something else? I have the same comment for line 148: "*The translational speed of all cyclone tracks is maintained at about 13 km/hour*", and in a few other places throughout the manuscript.

**Results and Discussion**
Line 218, 251, 264: if you want to include all of the values for the differences in MPS between parallel tracks, I suggest summarising them in a table rather than listing all of the values in the text. Consider a table also for the results in the text in Section 5.4.

Line 231, "*These angles bear a significant effect on the generation of the MPS as shown in the tables.*": here, I would suggest briefly mentioning the range of values of the MPS in the tables, to summarise the result, so that your reader doesn't have to open a separate document to have an overview of the result.

Line 270: Figure 4b is referred to but there isn't one.

Line 353: please include a reference to the data used for the storm tracks of historical cyclones.

Line 412: you mention that there is "good agreement" between the model and observations but there is no use of a statistical technique to confirm this. I'd suggest using RMSE here. Moreover, although you have explained in the text why the model might be underestimating the surge at landfall, you haven't discussed why the model overestimates the peak tide on other days. By eye, it looks like the model overestimates the high tide on 01/12 by at least 10cm. Why is this, and how could this bias influence biases on the day of the cyclone landfall? I'm not sure you can say the model is in good agreement with the amplitude of the tide without at least a discussion about where this isn't the case and why.

**Conclusions**
You have summarised your results and discussed their importance but you haven't mentioned their limitations. A discussion of the limitations of the study, any drawbacks to the methods used, and the future outlook for this topic would be appreciated. For example, how does the lack of

tides in the first two sets of experiments limit the study? How could this be accounted for in a future study on this topic?

**Figures**
Figure 3: A table or another type of 2D plot, or a combination like Fig. 4 could be a better way to show this result, showing each combination of T values and coastline shapes. The caption also needs more detail: from the caption alone I should be able to understand what the figure is showing, but here I need to read the text to understand the difference between i-ii and iii-iv.

Figures 8, 9 and 10: please move the text onto the land so it doesn't overlap with the data. Additionally, the colour scale should be changed since the highest values are not used (as far as I can see). Moreover, for these figures, please include the coordinates of the longitudes and latitudes of the region used.

Figure 11: Figure 11i would be improved by a change in the colours used: since there is no place with a sea level below ~0.5m, change the colour range to allow for better visualisation of the higher sea levels.

Figures 5, 11ii: some gridlines within the plot (as in Figure 6) would allow for easier reading of the values.

---

## Author Comment (AC1)

**Authors' Response to the Reviewers' Comments**

**Manuscript No.** egusphere-2024-2985

**Title of Paper:** "Investigation of complex coastline geometry impact on the evolution of storm surges along the east coast of India: A sensitivity study using a numerical model"

**The reply to the comments is given below:**

**Reviewer 1**

The work analyses the interaction between the shape of the coastline (convex/concave) and the trajectories of the cyclones in determining the exacerbation of the related storm surges. The modelling approach followed is robust and well-conceived, first with idealized test cases (regular coastline shapes) and straight cyclone tracks, then with real coastline and idealized cyclone tracks. Last a real case. My major remark is about this real case. I would suggest the author to move this case in the section 4 or at the beginning of section 5 to serve as validation of model setup which is then used for the idealized test cases and their discussion.

In the Results and Discussion, I would suggest summarizing the values of the MPS in tables, in order to make easier the reading of the text. The figures in my opinion need some small adjustments, which are suggested along the manuscript. Several other comments are highlighted along the text, in the attached PDF.

The manuscript is well referenced, well written and mostly easy to read and understand. For these reasons I recommend the manuscript to be eligible for publication after minor revision.

*Reply:* **Thank you for your encouraging comments. Each comment is addressed as given below:**

**Comment:** Line no. 59: Vulnerability is an intrinsic characteristic of an area with a certain coastline. Maybe it is the shape of the coastline which determines its vulnerability. Please explain better or review the sentence.

*Reply:* **As suggested by the reviewer, the shape of the coastline determines its vulnerability. The sentence is properly modified in the revised manuscript. Refer line no 59.**

**Comment:** Line no. 66: It is not clear to me what a cyclonic approach angle is. Do you mean a cyclone approaching parallel to a concave coastline?

*Reply:* **Approach angle of the cyclone refers to the angle making between the tangent drawn at the landfall location and the cyclone track measured clockwise. For clarity, this is included in the revised manuscript. Refer line no 66-67.**

**Comment:** Line no. 79: who is it? Maybe it is emphasized?

*Reply:* **Corrected. Changed to "emphasized" in the revised manuscript. Refer line no 80.**

**Comment:** Line no. 93: It would be helpful to show in a map the trajectory of the cyclone from the formation to the landfall.

*Reply:* **As suggested, the trajectory of the cyclone track is added and shown in Figure 1. Refer line no 97.**

**Comment:** Line no. 114: Please explain, without too many details, what is the hybrid formulation and the reason why there a minimum value of the drag coefficient (stability reasons?). Please, add units for the drag coeff. Ramp function and weighting factor: please explain what is and what is used for.

*Reply:* **As mentioned in** Luettich et al. (1992), **the hybrid bottom friction formulation is given as: FFACTOR = CF\*[1+(HBREAK/H)\*Exp(FTHETA)]\*Exp(FGAMMA/FTHETA) for (H < HBREAK). Otherwise, the hybrid friction formulation reverts to a standard quadratic formulation, in which FFACTOR=CF. Here, the value of CF is chosen as 0.0015 for model stability. Generalized Wave-Continuity Equation (GWCE) weighing factor (tau0) weights the relative contribution of the primitive and wave portions of the GWCE. The Ramp function simply scales the applied forcing, with no units, varies from 0 (no forcing) to 1 (full forcing).**

**In the above reference, detailed formulation for the hybrid friction is mentioned. The drag coefficient, Ramp function and weighing factor are having no units. This sentence is included in the revised text. Refer line no 119-125.**

*Reference:* *Luettich Jr, R. A., Westerink, J. J., and Scheffner, N. W. (1992), ADCIRC: An advanced three-dimensional circulation model for shelves, coasts, and estuaries. Report 1. Theory and methodology of ADCIRC-2DDI and ADCIRC-3DL (No. CERC-TR-DRP-92-6), COASTAL ENGINEERING RESEARCH CENTER VICKSBURG MS.*

**Comment:** Line no. 122: load is not appropriate: action or stress would be better.

*Reply:* **Corrected, changed it to stress in the revised manuscript. Refer line no 130.**

**Comment:** Line no. 124: work out is not clear.

*Reply:* **To make it more meaningful, the word "work out" is replaced by "compute" in the revised manuscript. Refer line no 132.**

**Comment:** Line no. 192: Coriolis?

*Reply:* **It is not Coriolis? The sentence is modified as "It is noticed that the trend is seen linearly decreasing with the increase in curvature of the domain from the track $T_{-2}$ to $T_2$."in the revised manuscript appropriately. Refer line no 227.**

**Comment:** Line no. 210: I do not see this increase. I see that MPS increases from T2 to T0. Between T-2 and T0 it is almost the same.

*Reply:* **Yes. there is no much change in MPS from CC1 to CC4 from the tracks T-2 to T0, However, the values increase for the more curvature shapes of CC5 and CC6. Refer Fig.5**

**Comment:** Line no. 231: All the analyses refer to the MPS, which is fine for me. I am also wondering on the average value of the SS along the coastline, and eventually its persistence. In other words, the combination which has the highest MPS, does it produce also the highest SS along a certain coastal area? A SS can be dangerous either for its MPS but also if it is widespread with high values along the coast.

*Reply:* **It is true that the MPS includes highest SS along the coastal area. Also, true that SS can be dangerous for its MPS as well as widespread. However, the widespread of the peak surge along the coast increases with the curvature. The maximum spread is seen with the highest curvature.**

**Comment:** Line no. 248: In Figure 5 it is not clear the time when MPS occurs. I guess there should be a MPS for each T. The black dot is the time of the landfall, but not the time of the MPS. It would be interesting to have a vertical line indicating it.

*Reply: We have not shown the time of occurrence of MPS in the figure. It is just 1 or 1.5 hours before or after the landfall time based on the curvature of the coast. As suggested, we replaced the black dot with a vertical black line. Refer: Figure 5 (Revised Fig. 7).*

**Comment:** Line no. 254: This is not evident.

*Reply: As this line is little ambiguous, the sentence is appropriately modified in the revised text. The revised sentence is "The onshore wind decreases and alongshore wind increases from track $T_{-2}$ to $T_2$ for CC6, which is consistent with the MPS" Refer line no. 298-299.*

**Comment:** Line no. 270: There is no Figure 4(b)

*Reply: Corrected, it is Figure 4. In the revised manuscript it is Figure 6. Refer line no. 309.*

**Comment:** Line no. 278: This sentence is not clear. Please revise. I understand it is a matter of extension of the domain: if this is true, then the resulting discussion is weekend by this issue.

*Reply: This sentence is modified suitably for more clarity. This can be attributed to reduction of available domain on the right side of any track in CC6 to drag water masses by the onshore winds towards the coast as we consider from $T_{-2}$ to $T_2$. Refer line no. 332-333.*

**Comment:** Line no. 281: In CC1 the surges first converge and then diverge. In CC6 they move parallel. I do not see only convergence. Please explain and discuss this. Sentence at line 283 is not enough.

*Reply: We agree with the reviewer's comment. To avoid confusion, we modified our discussion in the revised text. Refer line nos. 326-328 and 334-335.*

**Comment:** Line no. 302: I assume that the EB is plotted at the time of the landfall and not for instance at the time of the MPS. Why?

*Reply: Corrected. Yes, it is during the time of MPS not at the time of landfall. Refer line 354.*

**Comment:** Line no. 308: I would expect the funneling effect to concentrate the energy more in CC6 than in CC1, which has a higher peak value, but the other values are more spread and lower than those in CC6. The scale of the energy in Fig7 should be revised to show more clearly this.

*Reply:* *To avoid confusion, the sentence is suitably modified based on the simulations. For track $T_0$, the funneling effect is not seen in terms of surge generation (refer Table 1). The peak energy density per unit length is associated with the radius of maximum winds (Rmax) in either case. However, the spread/extent of the peak energy is seen more in CC6 compared to CC1 due to funneling effect. Corrected Fig.7 Refer line 359-361 and Fig. 9*

**Comment:** Line no. 335: Maybe northward would be better? Consider in the following paragraph to use north-south instead of left-right.

*Reply: Corrected in the revised manuscript. Refer line nos. 383,391,393,394,397 and 398.*

**Comment:** Line no. 340: It would be also interesting to see the location of the negative surge. Maybe this would be achieved with a different scale of diverging colors?

*Reply:* *The output file from the ADCIRC known as maxele gives only the maximum surge at each node during the whole cyclonic period. Some nodes may have the negative surge at some time, but it will eventually assign 0 m (highest value). Since the model stores only positive peak surges during the cyclone period, negative surges are not shown here. However, negative surges can be shown at a particular time as a snapshot. However, the Fig. 8 gives both maximum negative and positive surges along the coast with time.*

**Comment:** Line no. 341: Same as above

*Reply: Corrected in the revised manuscript. Refer line 397.*

**Comment:** Line no. 359: I would suggest adding north-center-south on the left side at each row of Figure9, and the angles at the top of the three columns. This would help the comprehension of the paragraph, which is a hard to follow.

*Reply: Corrected in the revised manuscript. Refer revised Fig. 10*

**Comment:** Line no. 374: Why there are names in red color? Please avoid as much as possible to overlap names over the colored map.

*Reply:* **Corrected in the revised manuscript. Refer revised Fig. 10, 11 and 12**

**Comment:** Line no. 388: Same comment as previous figure. Names could be smaller.

*Reply:* **Corrected in the revised manuscript. Refer revised Fig. 10, 11 and 12**

**Comment:** Line no. 395: Does the increase of Rmax affect the magnitude of the wind? Please comment this. A brief recap sentence at the end of the paragraph would be appreciable: something like. the closer the track, the higher the PS, The larger the radius, the larger the PS

*Reply:* **The increase of Rmax does not affect the magnitude of the wind but modifies the cyclonic horizontal wind distribution. It is expected higher surges are generated at far off coastal places from the landfall point as the Rmax increases. This is included appropriately. Refer line no. 452-454.**

**Comment:** Line no. 399: Is this test case simulated with the same model/domain configuration of the idealized cases? I guess so. If this is true, I would consider moving this paragraph before the idealized test cases. The reason is that this real case could provide the proof of the model validation. Once the model is validated can be used for the idealized cases.

*Reply:* **Corrected in the revised manuscript. Moved this section to the start of the Results and Discussion section. Refer to section 5.1**

**Comment:** Line no. 401: Which is the amplitude of the spring tide? It would be interesting to compare the local tide at the time of the cyclone with the local spring tide.

*Reply:* **The cyclone was passing the station "Krishnapatnam" at the time of spring tide and its value was 0.5m. It means local tide and the spring tide was the same at the time of cyclone. This sentence is modified to include this fact. Refer Line no. 198.**

**Comment:** Line no. 403: I would prefer west and east.

*Reply:* **Corrected in the revised manuscript. Refer revised line 200.**

**Comment:** Line no. 412: Please calculate some statistics such as BIAS, MAE, RMSD for the time series in Fig11(ii). Those values can be inserted also in the graph.

*Reply: Corrected and included in the figure as well in the revised manuscript.*

*"The computed correlation coefficient, mean absolute error (MAE), mean square error (MSE), mean bias error (MBE) and root mean square error (RMSE) are 0.91, 0.146 m, 0.032 m, 0.04 m and 0.16 m respectively."*

*Refer to line no 208-209 and revised Fig. 4 (ii).*

**Comment:** Line no. 421: Which is the time of this map? The landfall time? Please specify.

*Reply: No, it does not signify storm tide at the time of landfall. This is the plot of maximum storm tides generated during the entire cyclonic period. Refer line no 194 and Fig. 4(i).*

**Comment:** Line no. 451: Is the authors still discussing the Exps1? I guess no because of the line break. So, if the Exps2 are going to be introduced, please review this sentence.

*Reply: Yes, here we are discussing Exp2. The corrected sentence is "Exp2 is carried out with the real coastline of having complex geometry covering from Kothapatnam to Antervedi, which is one of the most prominent cyclone prone regions along the east coast of India". Since in the revised manuscript, previous Exp2 has become Exp3. Refer to line 487.*

---

## Author Comment (AC2)

**Authors' Response to the Reviewers' Comments**

**Manuscript No.** egusphere-2024-2985

**Title of Paper:** "Investigation of complex coastline geometry impact on the evolution of storm surges along the east coast of India: A sensitivity study using a numerical model"

**The reply to the comments given is below:**

**Reviewer 2**

Review of the manuscript "***Investigation of complex coastline geometry impact on the evolution of storm surges along the east coast of India: A sensitivity study using a numerical model***" by Tiwari et al., 2024.

**Overview:**

This paper discusses the effects of coastline shape and approach angle of a storm on peak storm surge, first using an idealised model, then expanding the analysis to the context of the coastline of India, and finally validates the model using Storm Michaung, a recent cyclone in the region. Previous literature on this topic analyses the effects of concave and convex coastlines on storm surge, while other papers in the field analyse the approach angle of the storm on the peak storm surge. This manuscript takes these analyses further by also considering the sharpness of the curvature of the coastline and combining this with varying the approach angle.

In general, this manuscript addresses some interesting questions regarding the effects of coastal geometry on storm surges, using both idealised and "real world" models, and presents some novel results along with results that corroborate with those in the aforementioned literature. It should be improved in its discussion of the results, with a more in-depth analysis in the text highlighting how the results fit into the context of the field. The presentation of some of the results also needs improvement.

Overall, I recommend that this manuscript undergoes major revisions regarding the text in the results and discussion sections. Although the results are generally satisfactory and I do not suggest new analysis, the writing needs some significant improvements. I hope this is not disheartening since I believe that there are some really interesting results in the paper which are worthy of

publication, but I feel that the work needed on the text in some sections will likely require more time than the deadline for minor revisions would allow.

*Reply: Thank you for your encouraging comments. We have modified the manuscript accordingly and a detailed response to each comment is given below.*

**General comments:**

**Comment:** In the introduction, you refer to several papers which assess the impact of coastal geometry on storm surges, but these papers are not mentioned again after the initial literature review. You should refer back to these papers in your results and conclusion, to discuss how your results compare to theirs, whether or not they are consistent with them and why, etc. It would also be interesting to understand better from the introduction how your study is unique from the others discussed. I don't understand well from reading the manuscript which of your results are original and if some of your results mostly replicate results from other studies, or how your analysis fits into the wider context of extreme sea level events in the region. This is an important context that is missing.

*Reply: In the introduction, we have mentioned the limitations of the prior studies. We have also mentioned the objective of our study which is different from the previous studies. Refer line no. 76-83. In the conclusion part, we have mentioned how our results align and are unique from other studies. Refer line nos.520-527.*

**Comment:** In the results sections, instead of referring to the left/right of the storm, use west/east. Moreover, many results are listed in the text rather than presented in the form of tables or figures, which makes them much more difficult to parse. Results that contain lists of more than a couple of values should be presented in tables or figures.

*Reply: Corrected; replaced the left/right to west/east. We have included the tables as suggested. Please refer Table nos.1-5.*

**Comment:** A general remark for the figures is to consider different colour map choices for contour plots rather than the rainbow/'jet' map chosen in several figures. Please check the Ocean Science figure guidelines for more information on choosing the best colour maps: https://www.ocean-science.net/submission.html#figurestables.

*Reply: Corrected. We changed the figures having rainbow/jet colormap according to guidelines. We have gone through the colormap as suggested in the link given above.*

**Comment:** In the figures you use a black dot in several time series to represent the landfall time. A vertical line would be much clearer. This is the case for figures 5, 6, and 11b.

*Reply: Corrected, changed the blacked dot to vertical line at the time of landfall. Please refer revised. Fig. nos. 5(ii), 8 and 9.*

**Specific comments:**

**Introduction**

**Comment:** The first paragraph of the introduction doesn't contain any references to literature. Perhaps you can refer to a book chapter or literature review that gives a more comprehensive overview of the topic of storm surges and coastline geometry. I see that you want to introduce the topic and highlight its importance, and I think this paragraph otherwise achieves this goal well, it just needs to cite the correct sources.

*Reply: Provided two appropriate references as suggested. Refer line no. 31 and 35.*

**Comment:** Line 76: add a reference to your existing Figure 1 when you describe the coastline of India.

*Reply: Provided a reference. Refer line no. 85.*

**Comment:** Line 77, "*Several cyclones in the past hit these coastal stretches with different approach angles. Some of the cyclones move parallel to these areas without making any landfall.*": please provide some examples of cyclones which have followed these paths, or references from literature to studies on these types of storms.

*Reply: Corrected. Refer line no. 86-87.*

**Synoptic History of the Michaung Cyclone**

**Comment:** I think that this section should be added to the introduction as a subsection; it's a very short paragraph to have a section to itself, and it is part of the general overview of the topic.

I recommend including a figure of Cyclone Michaung's entire track to accompany this description, labelling the significant dates and locations mentioned in the text.

*Reply: As suggested, moved this section as a subsection (section 1.1) of the "Introduction" and added a figure representing cyclone Michaung track. Refer line no. 103-113 and revised Fig.1.*

**Model Description**

**Comment:** Line 121, "*The sea level rises one cm for each hPa decrease in atmospheric pressure. The wind load on the sea surface leads to a notable fluctuation in sea level, influenced by the maximum wind speed and geographic distribution of winds*": please include an equation or a reference for these statements.

*Reply: Added the reference. Refer line no. 140.*

**Data and Methodology**

**Comment:** It would be useful to split this section into three subsections, one for the idealised experiments, the second for the real coastline experiments and a third for Cyclone Michaung. I would also find it useful to see a table summarising each set of experiments for reference, particularly the idealised experiments since there are lots of them.

*Reply: Modified. As suggested, divided the section into 3 subsections and added a table summarizing experiments (Section 3.1, 3.2 and 3.2). Refer line no. 148,160 and 203 and Table.1.*

**Comment:** Line 133, "*The bathymetry of the region also follows a concave-shaped coastline.*": could you clarify the wording of this sentence? I'm not sure whether you mean that the coastline is concave, or that in addition, the bathymetry is smooth and follows the same shape.

*Reply: Yes, the bathymetry of this particular region follows the concave shape which aligns with the concave shape of the coastline.*

**Comment:** Line 135: **a** and **b** are not defined in the manuscript. Please include some equations or a schematic figure to define them. I would suggest moving Figure S1 to the main manuscript since it shows **a** and **b** as well as being referred to later in your text.

*Reply: We have already defined the **a** and **b**. Refer line no. 173-175. For more information we included Figure S1 to the main manuscript as Fig. 4, along with the equation. Refer line no. 191-200.*

**Comment:** Line 146, "*The bathymetry of the domain is generated using a constant continental shelf width of about 50 km*": is it 50km exactly or something else? I have the same comment for line 148: "*The translational speed of all cyclone tracks is maintained at about 13 km/hour*", and in a few other places throughout the manuscript.

*Reply: Yes, it is exactly 50 km, which is the average continental shelf bathymetry over the region shown in Fig.2 having **a** = 25 km and **b** = 37.5 km. The translation speed of the idealized cyclone used in the study is 13km/hour everywhere in Exp1 and Exp2 (Refer: Revised Exp2 and Exp3).*

**Results and Discussion**

**Comment:** Line 218, 251, 264: if you want to include all of the values for the differences in MPS between parallel tracks, I suggest summarising them in a table rather than listing all of the values in the text. Consider a table also for the results in the text in Section 5.4.

*Reply: Since we have included Table 2 and Table 3, we are not including tables showing the difference in MPS. We have included Table. 4 and 5 for summarizing the onshore and alongshore winds and energy density per unit length.*

**Comment:** Line 231, "*These angles bear a significant effect on the generation of the MPS as shown in the tables.*": here, I would suggest briefly mentioning the range of values of the MPS in the tables, to summarise the result, so that your reader doesn't have to open a separate document to have an overview of the result.

*Reply: Yes, we have included Table 2 and Table 3 (previously in the supplementary) showing MPS for all the shapes and tracks along with the tangent angle.*

**Comment:** Line 270: Figure 4b is referred to but there isn't one.

*Reply: Corrected; it is Fig.4 (revised Fig.7); by mistake, it referred to 4(b). Refer line no. 331.*

**Comment:** Line 353: please include a reference to the data used for the storm tracks of historical cyclones.

*Reply: Corrected. Added the reference. Refer line no.429.*

**Comment:** Line 412: you mention that there is "good agreement" between the model and observations but there is no use of a statistical technique to confirm this. I'd suggest using RMSE here. Moreover, although you have explained in the text why the model might be underestimating the surge at landfall, you haven't discussed why the model overestimates the peak tide on other days. By eye, it looks like the model overestimates the high tide on 01/12 by at least 10cm. Why is this, and how could this bias influence biases on the day of the cyclone landfall? I'm not sure you can say the model is in good agreement with the amplitude of the tide without at least a discussion about where this isn't the case and why.

*Reply: Corrected. Included RMSE, MAE, MSE and MBE in the text as well as in the Figure. Refer line no. 234-235. The authors agree with the observation of the reviewer regarding the model simulates overestimates initially and underestimates later. However, the difference between model and observation is about 10cm, which is small. Hence, we changed the statement as follows.*

*"…….at Krishnapatnam is qualitatively in agreement with that of……."*
*Refer line no. 233*

**Conclusions**

**Comment:** You have summarised your results and discussed their importance, but you haven't mentioned their limitations. A discussion of the limitations of the study, any drawbacks to the methods used, and the future outlook for this topic would be appreciated. For example, how does the lack of tides in the first two sets of experiments limit the study? How could this be accounted for in a future study on this topic?

*Reply: Corrected. Included the limitations of the study. Refer line no. 528-536.*

**Figures**

**Comment:** Figure 3: A table or another type of 2D plot, or a combination like Fig. 4 could be a better way to show this result, showing each combination of T values and coastline shapes. The

caption also needs more detail: from the caption alone. I should be able to understand what the figure is showing, but here I need to read the text to understand the difference between i-ii and iii-iv.

*Reply: As suggested, included the Tables 2 and 3 to show MPS. Improved the caption of Fig.3. Refer revised Fig.6*

**Comment:** Figures 8, 9 and 10: please move the text onto the land so it doesn't overlap with the data. Additionally, the colour scale should be changed since the highest values are not used (as far as I can see). Moreover, for these figures, please include the coordinates of the longitudes and latitudes of the region used.

*Reply: Corrected. Please refer Figs. 11,12 and 13.*

**Comment:** Figure 11: Figure 11i would be improved by a change in the colours used: since there is no place with a sea level below ~0.5m, change the colour range to allow for better visualisation of the higher sea levels. Figures 5, 11ii: some gridlines within the plot (as in Figure 6) would allow for easier reading of the values.

*Reply: Changed the colormap of figure 11 (i). The range of colorbar is 0-1 m. Incorporated gridlines in the figures. Refer revised fig.5 (i).*